# Control of lysosome function by the GTPase-activating protein TBC1D9B and its binding partner TMEM55B

Valentin Duhay [1,10], Miaomiao Tian [2,10], Klaudia Kosieradzka [2], Michael Ebner [2,9], Wen-Ting Lo [3], Michael Krauss [2], Henner-Linus Sprengel[1], Matthias Voss [1], Mara Riechmann [1], Jeffrey N. Savas [4], Michael Schwake [5], Volker Haucke [2,6,7,11] ✉ & Markus Damme [1,8,11] ✉

Lysosomes are highly dynamic organelles that serve antagonistic functions as terminal catabolic stations for the degradation of macromolecules and as central metabolic decision centers for anabolic growth signaling. Lysosome dysfunction is implicated in various human diseases. The physiological roles of lysosomes are linked to the control of lysosome position and dynamics via the activity of the kinesin-activating small GTPase ARL8. How the activity of ARL8 is regulated remains poorly understood. Here, we identify the GTPase-activating Tre-2/Bub2/Cdc16 (TBC) domain protein TBC1D9B as a critical negative regulator of ARL8B function. We demonstrate that TBC1D9B is associated with the lysosomal membrane protein TMEM55B, directly binds to ARL8B-GTP, and stimulates its GTPase activity. Knockout of *TBC1D9B* or its binding partner *TMEM55B* causes lysosome dispersion, defective autophagic flux, and impairs the adaptive degradative response of cells to limiting nutrient supply. These lysosomal phenotypes of TBC1D9B loss are occluded by concomitant depletion of ARL8 in cells. Collectively, our data unravel a key role for TBC1D9B in controlling lysosome function by serving as a negative regulator of ARL8 activity.

Lysosomes are degradative organelles that mediate the hydrolytic breakdown of macromolecules delivered to their acidic lumen either by endocytosis or via autophagy through membrane fusion processes[1–3]. Besides their central importance for the catabolic turnover of macromolecules, lysosomes also fulfill a critical anabolic function by serving as a recruitment platform for the multisubunit nutrient-sensing and signaling mechanistic target of rapamycin complex 1 (mTORC1), a major driver of anabolic programs, such as protein and lipid biosynthesis[4,5]. For example, amino acids derived from lysosomal proteolysis are exported from the lysosome lumen to provide building blocks for protein synthesis while concomitantly promoting the activation of mTORC1. Active mTORC1 represses catabolic programs, such as autophagy and lysosomal gene expression via transcription factor EB (TFEB) and its relatives[1–3,6].

¹Institute of Biochemistry, Christian-Albrechts-University Kiel, Kiel, Germany. ²Leibniz-Forschungsinstitut für Molekulare Pharmakologie (FMP), Berlin, Germany. ³Graduate Institute of Biochemistry, College of Life Sciences, National Chung Hsing University, Taichung City, Taiwan. ⁴Ken and Ruth Davee Department of Neurology, Northwestern University Feinberg School of Medicine, Chicago, IL, USA. ⁵Biochemistry III/Faculty of Chemistry, Bielefeld University, Bielefeld, Germany. ⁶Department of Biology, Chemistry, Pharmacy, Freie Universität Berlin, Berlin, Germany. ⁷Charité-Universitätsmedizin Berlin, Berlin, Germany. ⁸Faculty of Chemistry, Biochemistry I (BCI), University of Bielefeld, Bielefeld, Germany. ⁹Present address: Institute of Molecular Biochemistry, Biocenter, Medical University of Innsbruck, Innsbruck, Austria. ¹⁰These authors contributed equally: Valentin Duhay, Miaomiao Tian. ¹¹These authors jointly supervised this work: Markus Damme, Volker Haucke. ✉e-mail: haucke@fmp-berlin.de; markus.damme@uni-bielefeld.de

These janus-faced activities of lysosomes as catabolic endpoints and decision-making centers in metabolic signaling are intimately linked to lysosome dynamics and positioning[7–9]. Lysosomes can move bi-directionally along microtubule tracks via plus- (e.g., kinesin) and minus-end (e.g., dynein) directed motor proteins[9–12], and their dynamics are further regulated by membrane contact sites (MCS) with other organelles, in particular with the endoplasmic reticulum (ER)[12–14]. Nutrient deprivation (e.g., amino acid starvation) triggers the perinuclear clustering of lysosomes[7], the acidification of the lysosome lumen[15], and concomitant elevation of lysosomal proteolysis, as well as the dissociation of mTORC1 from the lysosome surface[5], resulting in the repression of mTORC1 signaling[16]. Conversely, in fed cells under conditions of active nutrient signaling, lysosome motility is promoted, and lysosomes become dispersed to the cell periphery via kinesin-based mechanisms[7,16]. A major driver of the anterograde motility of lysosomes is ARL8 (i.e., ARL8A and ARL8B in mammalian cells), an N-acetylated member of the ADP ribosylation factor family of small GTPases[10,17,18]. When bound to the lysosome surface, active ARL8-GTP directly or indirectly via association with the adaptor protein PLEKHM2/ SKIP recruits kinesins KIF1A[19,20], and KIF5A-C[10] to promote microtubule-dependent lysosome dispersal to the cell periphery. In addition to PLEKHM2/ SKIP and KIF1A, other known ARL8 effectors include the hetero-hexameric tethering complex HOPS[21,22] that mediates autophagosome-lysosome fusion and the RUN- and FYVE-domain-containing proteins RUFY1[23], RUFY3, and RUFY4[11,24]. Despite the essential role of ARL8 in controlling lysosome dynamics and function in cells and tissues[18,25], we know surprisingly little about the regulatory factors that control ARL8 localization and activity. It has been demonstrated that the recruitment of ARL8 to the surface of lysosomes and lysosome-related organelles is promoted by the evolutionarily conserved multisubunit BLOC-one-related complex (BORC)[26,27], but the underlying mechanism remains uncertain. Importantly, no specific factors (e.g., guanine nucleotide exchange factors (GEFs) and GTPase-activating proteins (GAPs)) that control the nucleotide status and, thereby, the activity of ARL8 have been identified.

Here, we fill this knowledge gap by identifying TBC1D9B, a poorly characterized endo-lysosomal member of the Tre-2/Bub2/Cdc16 (TBC) domain protein family[28], as a negative regulator of ARL8B function and an interaction partner of the multispanning lysosomal membrane protein TMEM55B[29–31]. TMEM55B is a 284-residue lysosomal protein and has been annotated as a phosphatidylinositol phosphatase (also named as PIP4P1)[32]. However, TMEM55B lacks sequence similarity to other lipid phosphatases, and in vitro activity as a lipid phosphatase could not be validated in a later study[33]. The crystal structure of TMEM55B also does not support its function as a phosphatase, but rather suggests a role as a protein-protein interaction platform[34]. In support of this, it was shown that TMEM55B promotes the perinuclear clustering of lysosomes via a mechanism that depends on the scaffold protein c-Jun N-terminal kinase (JNK)-interacting protein 4 (JIP4), which serves as an adaptor for the microtubule minus end-directed dynein-dynactin motor complex[30,31,35]. Other known interactors of TMEM55B include the phospho-RAB effector protein RILPL1[36] and members of the NEDD4-like family of ubiquitin E3 ligases[30].

Here, we use affinity purification-mass spectrometry and co-immunoprecipitation to identify TBC1D9B as a binding partner of TMEM55B and a negative regulator of ARL8B activity. Knockout (KO) of either *TMEM55B* or *TBC1D9B* causes the ARL8-dependent dispersion of lysosomes to the cell periphery. We further demonstrate that loss of TBC1D9B impairs starvation-induced autophagic flux and lysosomal proteolysis via a mechanism that depends on its GAP activity and on the presence of ARL8B. These findings unravel a crucial role for TBC1D9B in regulating lysosome function by repressing ARL8B activity.

## Results

### TMEM55B directly interacts with TBC1D9B

Given the roles of TMEM55B in lysosome positioning[31] and counteracting neurodegeneration[36] and oxidative stress[30], we searched for TMEM55B-interacting proteins. To this aim, we combined GFP nanobody-based affinity purification with mass spectrometric analyses (AP-MS) of material isolated from lysates of HEK293 cells expressing TMEM55B-eGFP or eGFP (Fig. 1a). Robust filtering of the candidate list for hits near undetectable in the negative controls revealed the bait TMEM55B as the top hit (Fig. 1b). JIP4, a previously identified interactor of TMEM55B[31], was also among the top hits, validating our screen. Proteins functioning in "Golgi organization", "localization to endoplasmic reticulum exit site", and notably, "endomembrane organization" were enriched in our candidate list according to Gene Ontology terms (Fig. 1c). When we compared our top candidate hits with TMEM55B interaction partners identified in the BioPlex 2.0 dataset[37], we found an overlap of four proteins of the top 25 (our AP-MS screen)/20 (BioPlex screen) hits, underscoring the high specificity and reproducibility of our screen. Among these proteins were JIP4, RNF213, BIRC6, and TBC1D9B (Fig. 1d). TBC1D9B was particularly interesting because of its putative function as a GTPase-activating protein (GAP) and its hypothesized role in endo-lysosomal trafficking, possibly by modulating the function of RAB proteins (i.e., RAB11A)[38] and/or related small GTPases. We bidirectionally validated the interaction between TMEM55B and TBC1D9B by immunoprecipitation in HeLa cells. TMEM55B-GFP was robustly co-precipitated with TBC1D9B-HA (Fig. 1e). Conversely, TBC1D9B-GFP co-purified with TMEM55B (Fig. 1e). No co-immunoprecipitation was observed for GFP alone. To further examine the physical interaction between TMEM55B and TBC1D9B, we used CRISPR/Cas9 genome editing to establish knock-in cells expressing TBC1D9B carrying a C-terminal HA-tag from its endogenous locus (Fig. 1f and Supplementary Fig. 1a, b), enabling detection with high-affinity antibodies. Tagged endogenous TMEM55B was specifically associated with TBC1D9B-HA (Fig. 1g). These data reveal complex formation between endogenous TMEM55B and TBC1D9B proteins. We then tested whether TMEM55B directly binds to TBC1D9B. To this aim, we expressed and purified the cytoplasmic domain of TMEM55B fused to GST and incubated it with recombinant TBC1D9B-eGFP produced in insect cells. In line with our co-immunoprecipitation results, we found that recombinant, purified TBC1D9B-eGFP directly associates with GST-TMEM55B but not with GST alone (Fig. 1h). Hence, TMEM55B directly interacts with TBC1D9B.

TBC1D9B is a multi-domain protein comprising two N-terminal "glutamic acid (E)-rich repeat accumulated in membrane" (GRAM) domains, a TBC domain, an EF-hand, and a putative "α-interaction domain" (AID) domain. We generated and analyzed a series of deletion constructs to dissect the TBC1D9B domain(s) responsible for complex formation with TMEM55B (Fig. 1i). We then conducted co-immunoprecipitation experiments from lysates of cells co-expressing TMEM55B-eGFP with individual HA-tagged TBC1D9B-domains (C-terminus (aa 810–1250), N-terminus (aa 1–450), TBC-domain (aa 450–810), TBC + EF-hand domain (aa 450–950), TBC + GRAM domain (aa 301–810)) (Fig. 1j). Robust co-precipitation of TBC1D9B-domains with TMEM55B-eGFP was observed for the C-terminus, the N-terminus, and the TBC + EF-hand domains, but not for the TBC-domain alone. Hence, complex formation between TBC1D9B and TMEM55B is mediated by multiple sites but does not involve the TBC domain that harbors its GAP activity.

Finally, to gain initial insight regarding the interaction site for TBC1D9B within TMEM55B, we examined the AlphaFold2 model of TMEM55B (Supplementary Fig. 1c). This analysis, together with co-immunoprecipitation experiments, revealed that TMEM55B associates with TBC1D9B via a folded cysteine-rich

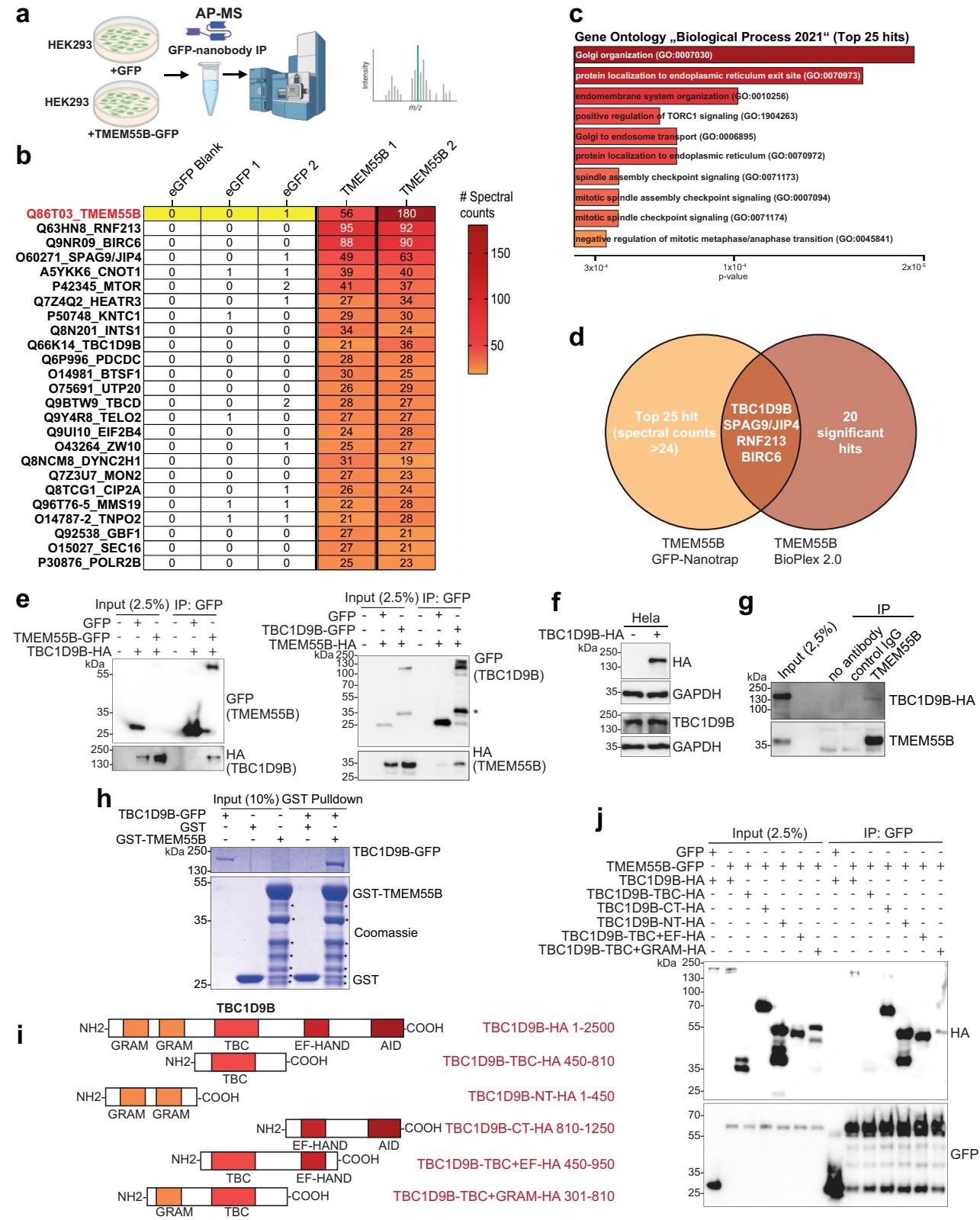

**i** TBC1D9B

TBC1D9B-HA 1-2500
TBC1D9B-TBC-HA 450-810
TBC1D9B-NT-HA 1-450
TBC1D9B-CT-HA 810-1250
TBC1D9B-TBC+EF-HA 450-950
TBC1D9B-TBC+GRAM-HA 301-810

domain starting from amino acid 65 (PDB: AF-Q86T03-F1-v4, AlphaFold), whereas the unstructured N-terminus of TMEM55B (i.e., amino acid residues 13–65) was dispensable for binding (Supplementary Fig. 1d, e).

Collectively, these data identify TBC1D9B as a binding partner of the integral lysosomal membrane protein TMEM55B.

## Loss of TBC1D9B or its binding partner TMEM55B causes lysosome dispersion

Previous studies have implicated TMEM55B in lysosome positioning, e.g., overexpression of TMEM55B causes lysosomes to coalesce in the cell center[31]. We, therefore, hypothesized that TBC1D9B might control lysosome position akin to its binding partner TMEM55B. To test a

**Fig. 1 | TBC1D9B interacts with TMEM55B. a** AP-MS strategy for the identification of TMEM55B-interacting proteins. Created in BioRender. Damme, M. (2026) https://BioRender.com/fjtc8jt. **b** A filtered list of TMEM55B-GFP candidate interacting proteins from HEK293 cells upon GFP-Nanotrap-based IP identified by MS, sorted by descending number of MS2 spectral counts in the two TMEM55B-eGFP/eGFP pulldown replicates. **c** Enriched Gene-Ontology terms of the top 25 TMEM55B-GFP interacting proteins sorted by decreasing *p*-value (Fisher's exact test, no adjustments for multiple comparisons). Gene-enrichment analysis was performed with the Enrichr tool (https://maayanlab.cloud/Enrichr/). **d** Venn diagram showing the overlap between the top 25 candidates identified in the AP-MS screen and the 20 significant hits in the "BioPlex 2.0" interactome[37] datasets. **e** Co-immunoprecipitation of TMEM55B and TBC1D9B in HeLa cells co-expressing TBC1D9B-HA and TMEM55B-eGFP (left panel) or TBC1D9B-eGFP and TMEM55B-HA (right panel) with a GFP-Nanotrap antibody followed by immunoblot with epitope-specific antibodies. GFP alone serves as a negative control. Cleaved GFP is marked with an asterisk (*). *n* = 3 independent experiments. **f** Immunoblot of CRISPR/Cas9-mediated endogenously HA-tagged TBC1D9B HeLa knockin cells with antibodies against HA, TBC1D9B, and GAPDH. *n* = 2 independent experiments. **g** Co-immunoprecipitation of TBC1D9B-HA knockin cells with an antibody against TMEM55B and TBC1D9B-HA. Lysates were subjected to TMEM55B antibody capture followed by immunoblot for HA and TMEM55B. Immunoprecipitations omitting the antibody and a non-related rabbit antibody serve as negative controls. *n* = 2 independent experiments. **h** In vitro pulldown experiment of recombinant purified TBC1D9B-eGFP incubated with GST-TMEM55B (cytoplasmic domain) bound to beads. Recombinant GST served as a negative control. GST, GST-TMEM55B, and TBC1D9B-eGFP were detected by Coomassie staining. TBC1D9B-GFP was loaded 10% of the total input. Asterisks (*) indicate partial proteolysis of GST-TMEM55B. *n* = 4 independent experiments. **i** Schematic representation of the domain structure of TBC1D9B and the truncated constructs used for IP experiments. **j** Co-immunoprecipitation of cells expressing full-length or truncated TBC1D9B-HA together with TMEM55B-eGFP. Lysates were subjected to GFP-Nanotrap antibody capture followed by immunoblot for HA and GFP. GFP-transfected cells serve as a negative control. *n* = 2 independent experiments.

possible physiological function of TBC1D9B in lysosome positioning, we used CRISPR/Cas9 to generate knockout (KO) HeLa cells lacking *TMEM55B* or *TBC1D9B* (Fig. 2a). Confocal imaging of wildtype *vs TMEM55B* KO cells stained for endogenous LAMP2 to mark lysosomes, paired with automated image analysis by OrgaMapper[39] revealed the accumulation of lysosomes in the cell periphery in *TMEM55B* KO cells, in agreement with previous work using siRNA[11,31] (Fig. 2b). Importantly, the loss of *TBC1D9B* also caused lysosomes to accumulate in the cell periphery, e.g., proximal to focal adhesion points, with a concomitant depletion of lysosomes within the perinuclear region (Fig. 2b–d), phenocopying *TMEM55B* KO in cells. TMEM55B and TBC1D9B KO cells, both displayed a trend towards increased cell size (Fig. 2b) and therefore we used cell size normalization in the quantification of lysosome positioning (see "Methods" section). Further, kinetic analysis of lysosome-movement by live-cell imaging of cells stained with LysoPrime Green revealed a significant increase in the fraction of motile lysosomes in both *TMEM55B* and *TBC1D9B* KO compared to wildtype cells (Fig. 2e). Moreover, the average speed of lysosome movement was increased in *TBC1D9B* KO cells, a phenotype that was rescued by re-expression of TBC1D9B-eGFP (Supplementary Fig. 2a). RAB11A-positive recycling endosomes were also partially redistributed to the cell periphery in *TBC1D9B* KO cells (Supplementary Fig. 2b), consistent with an earlier study and the known function of TBC1D9B as a GAP for RAB11A[38]. In contrast, the positioning of mitochondria (stained for cytochrome oxidase IV (Cox IV)), EEA1-positive early endosomes, and RAB8-positive endosomes was unaltered by the KO of either *TMEM55B* or *TBC1D9B* (Supplementary Fig. 2b). KO of *TBC1D9*, a TBC protein related to TBC1D9B, caused a mild accumulation of lysosomes in the cell periphery and a this phenotype was exacerbated by double loss of both TBC1D9 and TBC1D9B (*TBC1D9/TBC1D9B* DKO) (Supplementary Fig. 2c).

Given the physical association of TBC1D9B with lysosomal TMEM55B, we focused on the function of TBC1D9B at lysosomes. First, we examined the localization of TBC1D9B. TBC1D9B-HA expressed in *TBC1D9B* KO cells lacking the endogenous protein was found to partially colocalize with lysosomes marked by LAMP2 (Fig. 2f). Second, we tested whether the role of TBC1D9B in lysosome positioning and dynamics requires its GAP activity. To analyze this, we conducted rescue experiments by re-expressing wildtype TBC1D9B-HA or a GAP activity-defective point mutant with mutations in the arginine (R) and glutamine (Q) fingers[38,40] (i.e., R559A/Y592A/Q594A (TBC1D9B^RYQ/AAA)) (see also Fig. 4a) in *TBC1D9B* KO cells. Automated image analysis of the distribution of lysosomes marked by LAMP2 demonstrated that catalytically active wildtype TBC1D9B but not the inactive GAP mutant could rescue lysosome dispersion induced by loss of TBC1D9B (Fig. 2g, h).

## TBC1D9B and TMEM55B are required for the cellular response to starvation

Previous work has shown that the cellular response to starvation involves the perinuclear clustering of lysosomes and elevated lysosomal proteolysis, e.g., increased cathepsin activity[7,16]. To probe whether loss of TBC1D9B or TMEM55B affects the cellular starvation response, we monitored the distribution of lysosomes in cells subjected to starvation. As expected, starvation induced a marked relocalization of lysosomes towards the microtubule-organizing center in wildtype cells. In contrast, lysosomes maintained their largely peripheral distribution in *TMEM55B* or *TBC1D9B* KO cells (Fig. 3a, b). Next, we tested the consequences of the altered distribution of lysosomes in *TMEM55B*- and *TBC1D9B* KO cells on lysosomal proteolysis probed by SiR-Lyso, a pepstatin A-coupled silicon rhodamine derivative that binds cathepsin D. *TMEM55B*- and *TBC1D9B* KO cells tended to display a slight, albeit statistically insignificant increase in cathepsin D activity under fed steady-state conditions (Supplementary Fig. 3b). Importantly, *TMEM55B*- and *TBC1D9B* KO cells - unlike wildtype cells—failed to upregulate cathepsin D activity in response to starvation (Fig. 3c, d). We challenged these results from cathepsin D activity-based probes by assaying proteolytic cleavage of DQ-BSA, a quenched fluorophore coupled to BSA taken up by endocytosis that emits fluorescence upon lysosomal proteolysis. As seen for SiR-Lyso, *TMEM55B*- and *TBC1D9B* KO cells revealed a significant defect in lysosomal proteolysis in response to starvation (Supplementary Fig. 3a); no significant differences were observed in the uptake of fluorescently labeled dextran, excluding the possibility that the observed differences are due to reduced DQ-BSA endocytosis (Supplementary Fig. 3b).

To functionally probe starvation, we employed a recently developed pulse-chasable reporter processing assay for mammalian autophagic flux that is based on a HeLa cell line stably expressing Halo-LC3B. In this assay, the autolysosomal cleavage of Halo-LC3B to yield free Halo^ligand detected by in-gel fluorescence imaging, reflects autophagic flux[41]. We effectively depleted stable Halo-LC3B HeLa cells of endogenous TBC1D9B (Fig. 3e). We then subjected cells to starvation and monitored the ability of control *vs.* TBC1D9B-depleted cells to process Halo-LC3B to free Halo^ligand. Whereas control and TBC1D9B-depleted HeLa cells displayed very low levels of free Halo^ligand at steady state, TBC1D9B-knockdown cells suffered from a reduced ability to process Halo-LC3B to free Halo^ligand in response to starvation compared to control cells (Fig. 3e), indicative of reduced autophagic flux. Reduced autophagic flux was also reflected by the partial accumulation of the autophagy adaptor and substrate p62, taken as a surrogate for autophagic flux, in TBC1D9B-depleted cells (Supplementary Fig. 3c, d).

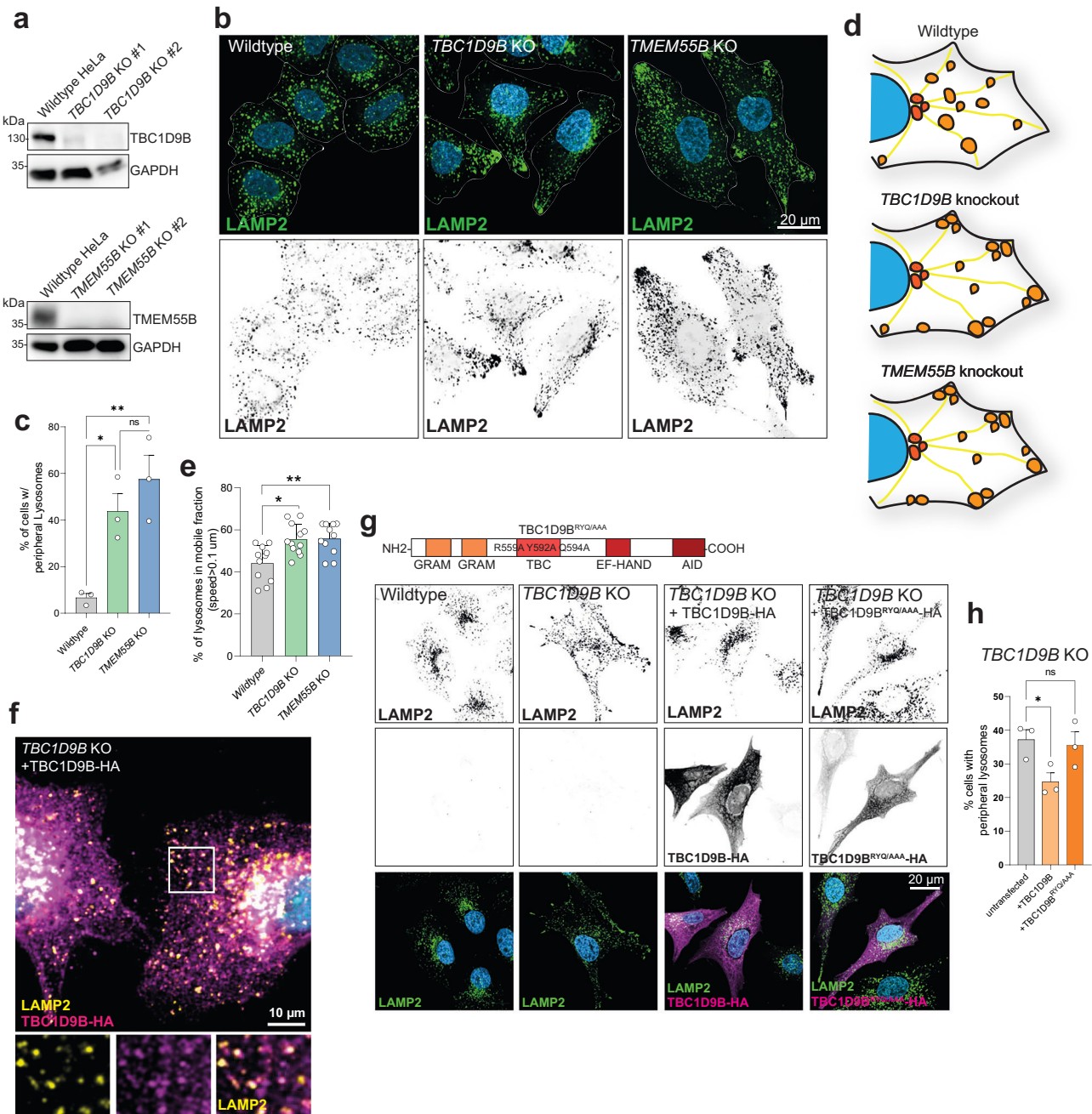

**Fig. 2 | Knockout of *TBC1D9B* or its binding partner *TMEM55B* causes lysosome dispersion. a** Immunoblot analysis of *TBC1D9B*-KO (upper panel) and *TMEM55B*-KO (lower panel) HeLa cells with antibodies against TBC1D9B and TMEM55B. GAPDH serves as a loading control. Two independent clones are shown. $n = 3$ independent experiments. **b** Lysosome dispersion in *TBC1D9B*-KO and *TMEM55B*-KO cells. Confocal microscopy images of *TBC1D9B*-KO and *TMEM55B*-KO cells stained for endogenous LAMP2 (green). Nuclei are stained with DAPI (blue). ns not significant; * = $p < 0.05$; ** = $p < 0.01$; wildtype vs. *TBC1D9B* KO $p = 0.0303$; wildtype vs. *TMEM55B* KO $p = 0.0073$. **c** Quantification of the number of peripheral lysosomes in wildtype, *TBC1D9B*-KO, and *TMEM55B*-KO cells, as determined by OrgaMapper. One-way ANOVA; ns not significant; * = $p < 0.05$; ** = $p < 0.01$. Mean ± SEM from $n = 3$ independent experiments. **d** Cartoon representation of the *TMEM55B/TBC1D9B*-KO effects on lysosomal positioning. The KO of *TMEM55B* or *TBC1D9B* leads to lysosome dispersion into the cell-periphery. **e** Quantification of the "percentage of lysosomes in the mobile fraction" of lysotracker red-stained

wildtype, *TMEM55B*-KO, and *TBC1D9B*-KO cells determined by confocal live-cell imaging microscopy. One-way ANOVA; ns not significant; * = $p < 0.05$; wildtype vs. *TBC1D9B* KO $p = 0.0225$; wildtype vs. *TMEM55B* KO $p = 0.0149$; mean ± SEM from $n = 1$ experiment with wildtype $n = 11$, *TBC1D9B* KO $n = 12$, *TMEM55B* KO $n = 11$ measurements. Each measurement represents the mean of all tracks of a cell. **f** TBC1D9B-HA partially colocalizes with LAMP2 in *TBC1D9B* KO cells. Confocal images of *TBC1D9B* KO HeLa cells stained for TBC1D9B-HA (magenta) and endogenous LAMP2 (yellow). **g** Rescue of defective lysosome position in *TBC1D9B*-KO cells requires TBC1D9B GAP activity. Immunofluorescence microscopy for LAMP2 (green) of wildtype and untransfected *TBC1D9B*-KO cells or *TBC1D9B*-KO cells re-expressing wildtype TBC1D9B-HA (magenta) or a GAP-defective mutant (TBC1D9B^RYQ/AAA-HA). Nuclei are stained with DAPI (blue). **h** Quantification of the number of peripheral lysosomes determined by Orga-Mapper. One-way ANOVA; ns not significant; * = $p < 0.05$; vs. full length $p = 0.0428$. Mean ± SEM from $n = 3$ independent experiments.

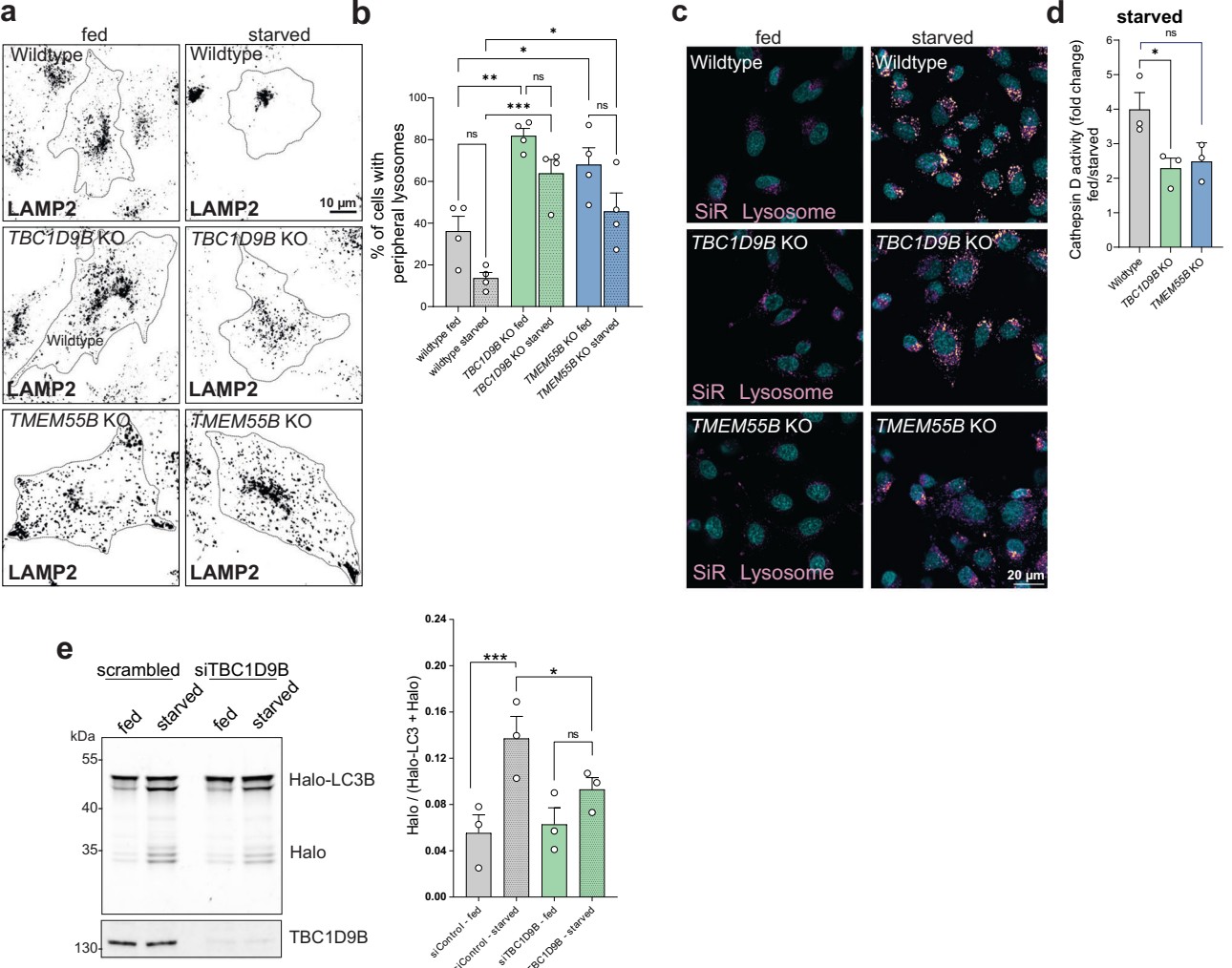

**Fig. 3 | TBC1D9B and TMEM55B are required for the cellular response to starvation. a, b** Loss of TBC1D9B or TMEM55B impairs starvation-induced perinuclear clustering of lysosomes. **a** Confocal images of fed and starved (6 h) wildtype, *TBC1D9B*-KO, and *TMEM55B*-KO HeLa cells stained for LAMP2. **b** Quantification of the number of peripheral lysosomes in the different genotypes and under the different conditions by OrgaMapper. One-way ANOVA; ns not significant; * = $p < 0.05$; ** = $p < 0.01$; *** = $p < 0.001$. Mean ± SEM from $n = 4$ independent experiments; wildtype fed vs. *TBC1D9B* KO fed $p = 0.03$; wildtype fed vs. *TMEM55B* KO fed $p = 0.0013$; wildtype starved vs. *TBC1D9B* KO starved $p = 0.029$; wildtype starved vs. *TMEM55B* KO starved $p = 0.00047$. **c, d** TBC1D9B and TMEM55B are required for the starvation-triggered induction of cathepsin D activity. **c** SiR-Lysosome staining of live fed- and starved cells (EBSS starved, 4 h) of the indicated genotypes. **d** Quantification of Cathepsin D activity in starved cells (relative to the unstarved/fed situation). One-way ANOVA; ns not significant; * = $p < 0.05$. Mean ± SEM from $n = 3$ independent experiments. **e** TBC1D9B depletion reduces autophagic flux in starved cells. (Left) immunoblotting and in-gel-fluorescence detection of total lysates of HeLa cells stably expressing Halo-LC3B upon treatment with the indicated siRNAs. (Right) quantification of Halo-LC3B cleavage. Halo band intensities were normalized by the sum of band intensities determined for Halo-LC3B and Halo. ($n = 3$ independent experiments). Mean ± SEM from $n = 3$ independent experiments. Statistics: two-way ANOVA, followed by Tukey´s multiple comparisons test. (siControl-fed vs. siControl-starved: $p = 0.002$; siTBC1D9B-fed vs siTBC1D9B-starved: $P = 0.2046$; siControl-fed vs siTBC1D9B-fed: $p = 0.9573$; siControl-starved vs siTBC1D9B-starved: $p = 0.0357$).

Collectively, these data show that TBC1D9B and its binding partner TMEM55B control lysosome positioning and, thereby, autophagic flux and the cellular response to starvation.

## Overexpression of TBC1D9B perturbs lysosomal positioning

Overexpression of TMEM55B causes perinuclear clustering of lysosomes[31]. We, therefore, tested the effect of TBC1D9B overexpression on lysosomal positioning. Ectopic expression of HA-tagged TBC1D9B in wildtype HeLa cells, followed by immunofluorescence staining, revealed a mostly cytoplasmic localization of TBC1D9B (Fig. 4a, b, d), in contrast to what was observed in *TBC1D9B* KO cells. Automated image analysis by OrgaMapper revealed the perinuclear clustering of lysosomes towards the cell center in cells overexpressing TBC1D9B (Fig. 4e), a phenotype similar, although less prominent, to

that elicited by overexpression of TMEM55B. In contrast, overexpression of GAP-defective TBC1D9B[RYQ/AAA] did not affect lysosome positioning (Fig. 4c), consistent with the hypothesis that TBC1D9B functions as a GAP protein to control lysosome dynamics. Wildtype and GAP-defective TBC1D9B[RYQ/AAA] showed a similar degree of co-localization with LAMP2 (Fig. 4d).

## TBC1D9B directly interacts with active ARL8B

The data presented thus far show that TBC1D9B controls lysosome positioning in various cell types by serving as a GAP protein. We tested the hypothesis that TBC1D9B might control lysosome position via its known role as a GAP for RAB11A. To this aim, we monitored lysosome position in HeLa cells depleted of endogenous RAB11. We found that knockdown of RAB11 alone had no effect on lysosome distribution.

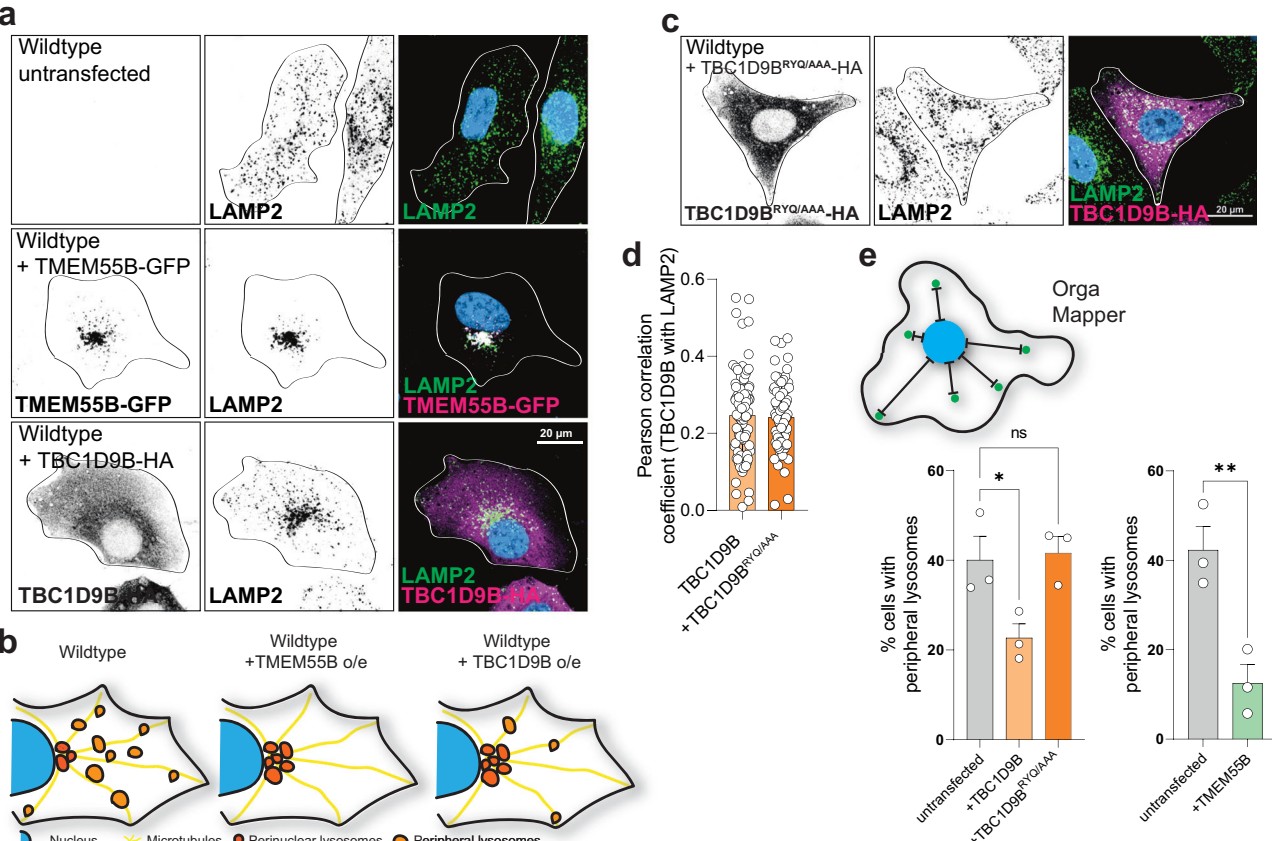

**Fig. 4 | TBC1D9B gain-of-function perturbs lysosome positioning. a** Confocal image of wildtype HeLa cells overexpressing TMEM55B (magenta) or TBC1D9B-HA (magenta) stained for endogenous LAMP2 (green). $n = 3$ independent experiments. **b** Cartoon representation of the effects of overexpression of TMEM55B or TBC1D9B on lysosome positioning. The overexpression of TMEM55B or TBC1D9B leads to perinuclear clustering of lysosomes. **c** Confocal images of wildtype HeLa cells overexpressing TBC1D9B^RYQ/AAA-HA stained for HA (magenta) and endogenous LAMP2 (green). $n = 3$ independent experiments. **d** Pearson correlation coefficient of TBC1D9B/TBC1D9B^RYQ/AAA-HA and LAMP2. Mean ± SD from pooled data of $n = 3$ independent experiments (TBC1D9B $n1 =$, $n2 = 27$, $n3 = 30$; TBC1D9B-RYQ/AAA $n1 = 28$, $n2 = 27$, $n3 = 21$). **e** Quantification of the number of peripheral lysosomes in each condition was determined by OrgaMapper. One-way ANOVA (left panel), t-test (right panel); ns not significant; * = $p < 0.05$; ** = $p < 0.01$. Mean ± SEM from $n = 3$ independent experiments; left panel: untransfected vs. full length $p = 0.043$; right panel; untransfected vs. transfected $p = 0.0021$.

Moreover, co-depletion of RAB11 and TBC1D9B in double knockdown cells did not show additive effects compared to the single loss of TBC1D9B (Supplementary Fig. 4a, b). These data suggest that the effects of TBC1D9B on lysosome distribution and function are not mediated via its known regulatory role on RAB11.

Previous work had identified ARL8 as an important regulator of axonal transport of lysosomes and synaptic vesicle precursor organelles in human induced pluripotent stem cell-derived neurons (iN)[20]. In a parallel independent effort to identify regulators of ARL8 function, we used lentiviral constructs to express ARL8B-TurboID or TurboID tagged with a nuclear export signal as a negative control in iN and analyzed proteins captured on paramagnetic streptavidin beads by semi-quantitative mass spectrometry (Fig. 5a). This analysis, in addition to known interactors of ARL8B, such as the kinesin adaptor PLEKHM2 (SKIP)[10], revealed the abundant presence of TBC1D9B (Fig. 5b). Given these findings and the fact that KO of either *TMEM55B* or *TBC1D9B* closely resembles ARL8 overexpression[17,42], we hypothesized that TBC1D9B might control lysosome positioning by acting on ARL8, i.e., a small GTPase for which no GAP protein has been identified.

To test this hypothesis, we analyzed the association of TBC1D9B with ARL8B. First, we assayed the ability of both proteins to undergo complex formation in HeLa cells co-expressing TBC1D9B-HA together with wildtype, active GTP-locked (Q75L), or inactive GDP-locked (T34N) variants of ARL8B-eGFP. Immunoblot analysis of material captured via GFP anotrap beads revealed avid TBC1D9B association with wildtype or constitutively active GTP-bound ARL8B (Q75L) but not with the inactive GDP-locked ARL8B variant (T34N) (Fig. 5c). A GAP-activity-defective TBC1D9B mutant (TBC1D9B^RYQ/AAA) retained the ability to interact with ARL8B-GFP (Q75L) (Supplementary Fig. 4c). Surprisingly, the ability of TBC1D9B to bind to ARL8 was restricted to the ARL8B isoform, as overexpressed wildtype or active GTP-locked (Q75L) ARL8A-eGFP failed to co-immunoprecipitate with TBC1D9B-HA (Supplementary Fig. 4d), despite > 90% sequence identity between both isoforms (Supplementary Fig. 4e). Next, we tested whether TBC1D9B and ARL8B directly interact. To test this, we conducted affinity chromatography experiments using recombinant glutathione S-transferase (GST)-tagged ARL8B (Q75L) or (T34N) as a bait and incubated these with recombinant TBC1D9-eGFP. Only the GTP-locked (active) ARL8B variant (Q75L) showed a robust direct interaction with TBC1D9B-eGFP (Fig. 5d and Supplementary Fig. 5a). Moreover, we found that TMEM55B, TBC1D9B, and ARL8B are capable of forming a tripartite complex in vitro. GST-TMEM55B, but not GST alone, was able to capture a complex between recombinant TBC1D9B-eGFP and native untagged ARL8B (Fig. 5e). Finally, we found that the overexpression of active GTP-locked ARL8B-eGFP (Q75L) in HeLa was sufficient to facilitate the recruitment of TBC1D9B to lysosomes in the cell periphery marked by LAMP2, while inactive ARL8B-GFP (T34N) failed to affect TBC1D9B localization (Fig. 5f).

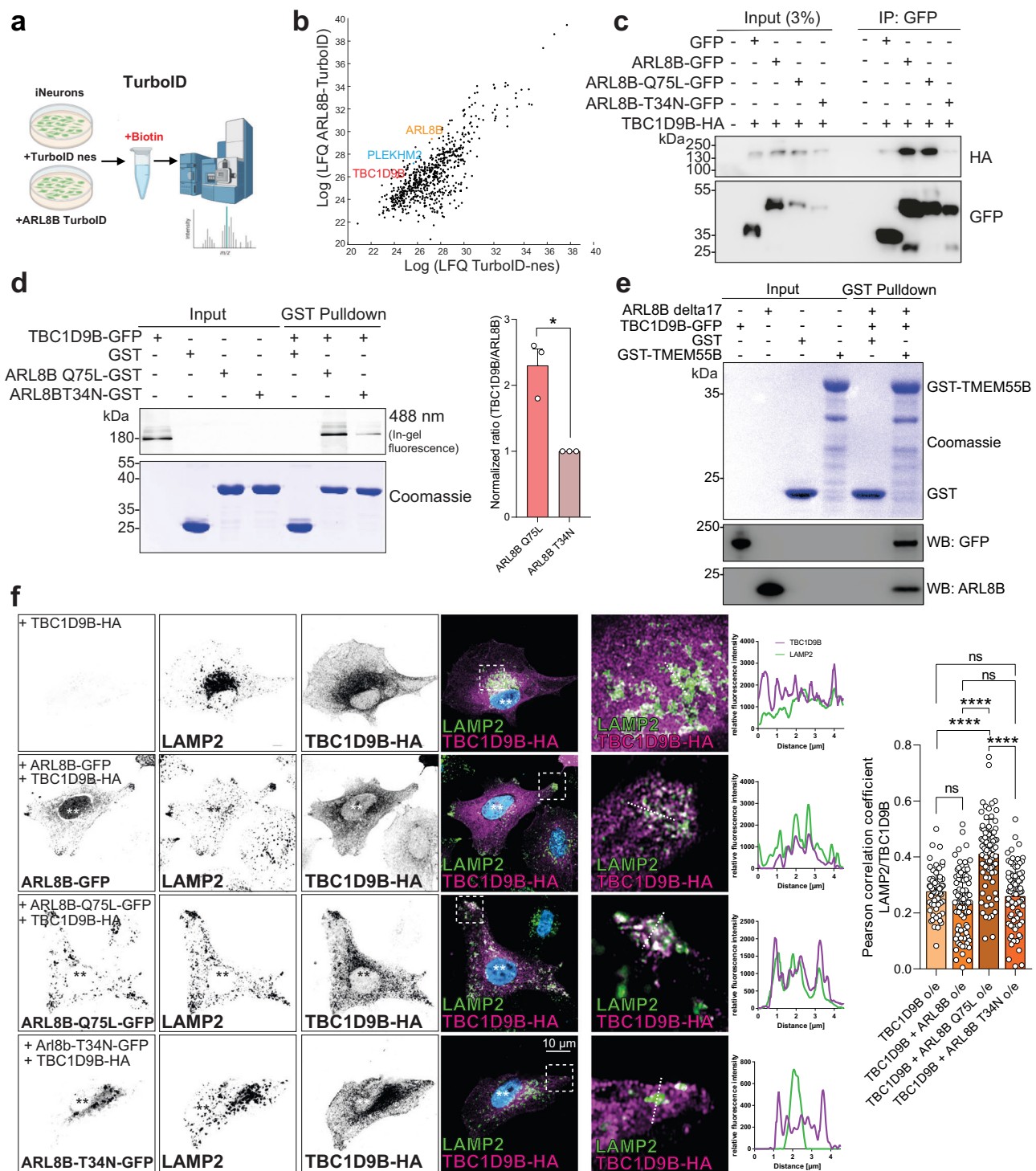

These results identify TBC1D9B as a specific direct binding partner of active ARL8B-GTP in vitro and in living cells and reveal the formation of a ternary interaction between TMEM55, TBC1D9B, and ARL8B.

## TBC1D9B facilitates GTP hydrolysis by ARL8B

Our collective results show that the TBC domain protein TBC1D9B directly binds to ARL8B in its GTP form, akin to the association of GAP proteins with their cognate active small GTPases. Moreover, KO of *TBC1D9B* (compare Fig. 3) or overexpression of ARL8[16,17,21,42] causes the peripheral dispersion of lysosomes, while conversely, the over-expression of TBC1D9B leads to perinuclear clustering of lysosomes

similar to the loss of ARL8B[16,21] (compare Fig. 4). Hence, TBC1D9B appears to act as a negative regulator of ARL8 function.

We used AlphaFold3 to obtain a structural model for the complex between TBC1D9B and ARL8B bound to GTP (Fig. 6a and Supplementary Fig. 5a). The interaction was predicted with very good confidence and revealed residues (e.g., R559, Y592, and Q594) critical for GAP activity[40] to contact the GTP binding pocket of ARL8B (Fig. 6b). Based on this analysis, we hypothesized that TBC1D9B may facilitate GTP hydrolysis by ARL8B. We tested this by purifying recombinant $His_{10}$-TBC1D9B wildtype or a GAP-inactive mutant (R559A, Y592A, and Q594A) from Sf21 insect cells (Supplementary Fig. 5c). Moreover, we expressed and purified recombinant GST-tagged RAB11A, a known

**Fig. 5 | TBC1D9B directly associates with active ARL8B-GTP in living cells.**
**a** TurboID strategy for the identification of TBC1D9B-interacting proteins in iPSC-derived iNeurons. Created in BioRender. Damme, M. (2026) https://BioRender.com/nh1fa4i. **b** Untargeted label-free quantitation (LFQ) of the ARL8B proximity interactome in iNeurons. **c** TBC1D9B specifically associates with active ARL8B-GTP. Immunoprecipitation using GFP-Nanotrap antibodies from lysates of HeLa cells expressing different ARL8B variants (wildtype, Q75L, T34N) and TBC1D9B-HA. Samples were analyzed by immunoblotting for GFP and HA. GFP-transfected cells served as a negative control. $n = 3$ independent experiments. **d** In vitro pulldown experiment of recombinant purified TBC1D9B-eGFP incubated with different GST-ARL8B variants (Q75L, T34N) bound to beads. Recombinant GST served as a negative control. GST/GST-fusion proteins were visualized by Coomassie staining (false-color blue) and TBC1D9B-GFP by in-gel GFP fluorescence. Quantification of the relative binding of TBC1D9B-GFP to ARL8B Q75L/ T34N. One-sample t-test; two-sided; $p = 0.0355$. Mean ± SEM from $n = 3$ independent experiments. **e** In vitro pulldown experiment of recombinant purified GST-TMEM55B/GST alone and TBC1D9B-eGFP incubated with recombinant ARL8B lacking amino acids 1–17 (ARL8B delta17). GST-TMEM55B/GST were bound to beads. GST/GST-TMEM55B-GST-fusion protein was visualized by Coomassie staining (false-color blue), TBC1D9B-GFP, and ARL8B delta17 by immunoblot. $n = 3$ independent experiments. **f** Confocal images of wildtype HeLa cells overexpressing TBC1D9B-HA alone or together with the indicated eGFP-tagged-ARL8B variants (wildtype, Q75L, T34N; white) stained with antibodies against HA (magenta) and LAMP2 (green). Nuclei are stained with DAPI (blue). A representative experiment from 3 replicates is shown for the immunofluorescence panel. Fluorescence intensity profiles are depicted for the indicated cross-sections. The Pearson correlation coefficient of TBC1D9B-HA and LAMP2 is depicted. Bar graph: One-way ANOVA; mean ± SEM from $n = 2$ independent experiments; ns not significant; **** $= p < 0.0001$; TBC1D9B o/e vs. TBC1D9B + ARL8B Q75L o/e $p = 2.48 \times 10^{-11}$; TBC1D9B + ARL8B o/e vs. TBC1D9B + ARL8B Q75L o/e $p = 9.6 \times 10^{-13}$; TBC1D9B + ARL8B T34N o/e vs. TBC1D9B + ARL8B Q75L o/e $p = 9.6 \times 10^{-13}$; TBC1D9B o/e $n1 = 29$, $n2 = 32$; TBC1D9B + ARL8B o/e $n1 = 47$, $n2 = 35$; TBC1D9B + ARL8B Q75L o/e $n1 = 51$, $n2 = 31$; TBC1D9B + ARL8B T34N o/e $n1 = 43$, $n2 = 39$.

---

substrate for TBC1D9B, as well as full-length GST-ARL8B (Supplementary Fig. 5b). We then conducted GDP Glo assays to analyze the ability of ARL8B ($6 \mu M$) to hydrolyze GTP in the absence or presence of a substoichiometric concentration ($0.5 \mu M$) of wildtype or GAP-defective mutant (i.e., TBC1D9B$^{RYQ/AAA}$) recombinant TBC1D9B. We found that active wildtype TBC1D9B stimulated GTP hydrolysis by ARL8B in a time-dependent manner, whereas the GAP-defective mutant was inactive (Fig. 6c). Recombinant wildtype or mutant TBC1D9B did not display any measurable GTP hydrolyzing activity on their own (Supplementary Fig. 5d). GST-ARL8B alone was also nearly inactive (Fig. 6c). Importantly, the GAP activity of TBC1D9B towards ARL8B was comparable to its ability to stimulate GTP hydrolysis by its known substrate RAB11A (Supplementary Fig. 5e, f). These data show that TBC1D9B facilitates GTP hydrolysis by ARL8B. It is conceivable that the GAP activity observed in our in vitro assays may be further stimulated in vivo by interactions of TBC1D9B with other proteins (e.g., TMEM55B), by membrane lipids, and/ or via posttranslational modifications that could alter the conformational state and, thereby, the activity of TBC1D9B.

A prediction from the model that TBC1D9B serves as a negative regulator of ARL8B function (e.g., by serving as a GAP) is that loss of TBC1D9B and, possibly, of its lysosomal binding partner TMEM55B should facilitate its recruitment to lysosomes. Indeed, we observed that loss of *TMEM55B* or *TBC1D9B* in KO cells boosted ARL8B-eGFP recruitment to lysosomes marked by LAMP2 (Fig. 6d).

Collectively, these findings identify TBC1D9B as a negative regulator of ARL8B function in vitro and in living cells.

### TBC1D9B controls lysosome function via ARL8B

Finally, we wanted to know whether its effects on ARL8B mediate the cellular effects of TBC1D9B loss. A prediction from our model is that the phenotypes caused by TBC1D9B loss should be occluded by the concomitant depletion of ARL8B. KO of *TBC1D9B* caused the dispersion of lysosomes to the cell periphery in control cells. In contrast, in cells depleted of ARL8B (Supplementary Fig. 5e), lysosomes remained perinuclearly clustered, irrespective of the presence or absence of TBC1D9B or its interactor TMEM55B (Fig. 7a, b). Hence, TBC1D9B controls lysosome positioning via a mechanism that depends on ARL8B.

Lysosomes are the degradative endpoint for autophagic cargo delivered to their lumen via fusion of autophagosomes (e.g., via the ARL8B-GTP effector HOPS[22]) under nutrient starvation conditions. Given the defects of TBC1D9B-depleted cells with respect to autophagic flux and the lysosomal starvation response (compare Fig. 3), we monitored the presence of LC3-positive autophagosomes in cells depleted of TBC1D9B, ARL8A/B, or both proteins. We observed that cellular depletion of TBC1D9B caused a significant reduction of autophagosomes marked by LC3 under nutrient starvation conditions. Conversely, we found autophagosomes to accumulate in double KO cells lacking *ARL8A* and *ARL8B* expression. This accumulation of LC3-positive autophagosomes persisted in *ARL8A/B* DKO cells depleted of TBC1D9B (Fig. 7c, d), suggesting that the partial depletion of LC3-containing autophagosomes in TBC1D9B-knockdown cells depends on the function of ARL8.

These collective data suggest that TBC1D9B controls lysosome function and autophagic flux via ARL8B (Fig. 7e).

## Discussion

Lysosome positioning and dynamics are crucial for nearly all aspects of lysosome function in health and disease, ranging from autophagy[43] and the cellular response to nutrient starvation[1–3] to lysosomal exocytosis during membrane repair[44], cancer[43,45], and neurodegeneration, e.g., Alzheimer's or Parkinson's disease, which are associated with the accumulation of stalled defective lysosomes in axonal varicosities[1,9,46]. Research over the past decade has unraveled ARL8 as a major controller of lysosome position and dynamics[17,21,47,48] in many, if not all, of the above processes.

Our results reported here suggest that TBC1D9B, likely via its association with the lysosomal membrane protein TMEM55B (Fig. 1), functions as a hitherto unknown negative regulator of ARL8B function. We present several convergent lines of evidence to support this proposal: first, we demonstrate that loss of TBC1D9B or TMEM55B in cells causes lysosomal dispersion to the cell periphery (Fig. 2) and counteracts the accumulation of LC3-positive autophagosomes via a mechanism that depends on ARL8 function (Fig. 7). Moreover, the phenotypes elicited by TBC1D9B loss with respect to lysosome position and dynamics resemble those induced by ARL8 gain-of-function[17,42]. Second, we find that loss of TBC1D9B or its interactor TMEM55B leads to the accumulation of active ARL8B on lysosomes (Fig. 6), consistent with TBC1D9B's role as a negative regulator of ARL8B activity, thereby facilitating the cellular response to starvation (see Fig. 3). Conversely, overexpression of TBC1D9B causes the perinuclear retention of lysosomes via a mechanism that depends on its GAP activity (Fig. 4). Third, we show that TBC1D9B directly and selectively associates with active ARL8B-GTP in living cells and in vitro (Fig. 5a–f). Fourth, we demonstrate that purified TBC1D9B, but not an inactive GAP mutant variant of it, facilitates nucleotide hydrolysis on recombinant ARL8B in vitro (Fig. 6). Based on our data, we, therefore, hypothesize that TBC1D9B may regulate the function of lysosomes and lysosome-related organelles (e.g., lytic granules in T cells, melanosomes, synaptic vesicle precursors in neurons)[20,25,49] in a variety of cell physiological and pathophysiological conditions beyond nutrient starvation, as demonstrated here (Figs. 3 and 7). Given its function as a selective repressor of ARL8B but not ARL8A activity, we expect

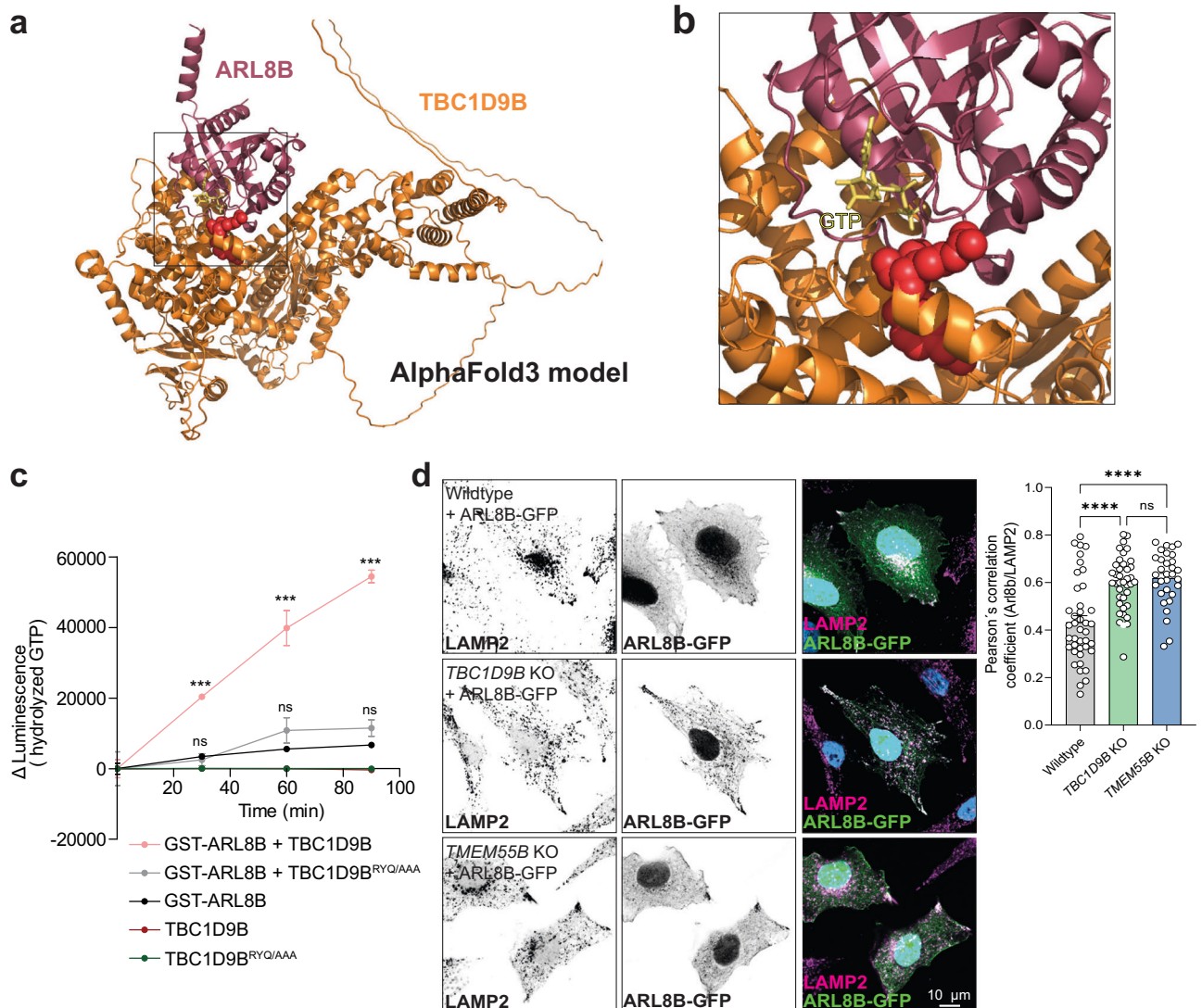

**Fig. 6 | TBC1D9B can promote nucleotide hydrolysis by ARL8B. a** Alphafold3 model for TBC1D9B (orange) bound to ARL8B (ruby-colored) in the presence of GTP (yellow sticks). The residues mutated in the TBC1D9B[RYQ/AAA] version are shown as spheres and highlighted in red. **b** Closeup view of theTBC1D9B-(GTP)-ARL8B interface. **c** GTPase activity of 6 μM GTP-bound GST-ARL8B was measured using the GTPase-Glo assay in the presence of 0.5 μM TBC1D9B or TBC1D9B[RYQ/AAA] at different incubation times. GTPase-Glo™ detects the remaining GTP upon the conversion to ATP; ATP is quantified by the addition of detection reagent and emitted luminescence. The change of luminescence indicates the amount of hydrolyzed GTP in the reaction. Data are presented as mean values ± SEM, $n = 3$ independent experiments. Two-way ANOVA, Bonferroni's multiple comparisons test; GST-ARL8B + TBC1D9B vs. GST-ARL8B, at 30 min, $p = 1.005 \times 10^{-5}$; at 60 min, $p < 1.4 \times 10^{-13}$; at 90 min, $p < 1 \times 10^{-14}$. **d** Confocal images of wildtype, *TMEM55B* knockout, and *TBC1D9B* knockout HeLa cells stained for LAMP2 and ARL8B-GFP. Left, Pearson correlation coefficient between LAMP2 and ARL8B. One-way ANOVA; mean ± SEM from $n = 1$ experiment with WT $n = 41$, *TBC1D9B* KO $n = 39$, *TMEM55B* KO $n = 31$; ns not significant; **** = $p < 0.0001$; Wildtype vs. *TBC1D9B* KO $p = 2.37 \times 10^{-6}$; Wildtype vs. *TMEM55B* KO $p = 5.47 \times 10^{-7}$.

phenotypes resulting from the organismal loss of TBC1D9B to be less severe compared to those reported for either loss of ARL8A and/ or ARL8B[20,50], or its activator BORC[51]. Future studies will be needed to assess the function of TBC1D9B in vivo.

How exactly complex formation between TBC1D9B and ARL8B in living cells and tissues is controlled remains unclear at present. One possibility is that the localization and/or activity of TBC1D9B is subject to regulation by other proteins and/or membrane lipids. The precise function and localization of TBC1D9B is also uncertain[38,52,53]. While one study showed a vesicular staining pattern in polarized MDCK cells for endogenous TBC1D9B (although the specificity of the antibody remains unknown)[38], in other studies, TBC1D9B was observed mainly in the cytoplasm[3] or in vesicles upon starvation and chloroquine treatment. In our experiments, we find overexpressed TBC1D9B to be partially colocalized with lysosomal LAMP2 in *TBC1D9B* KO cells

(Fig. 3a), whereas the majority of the protein remains in the cytoplasm, indicating transient binding of TBC1D9B to TMEM55B and ARL8B (i.e. a model supported by our in vitro data suggestive of a tripartite complex between TMEM55B, TBC1D9B, and ARL8B, see Fig. 5e). This is consistent with its function as a GAP for ARL8B, which requires only temporary contact with its small GTPase to facilitate nucleotide hydrolysis. While previous in vitro data showed GAP activity towards RAB8 and RAB11[38], our data add ARL8B as a further physiological substrate for TBC1D9B at lysosomes. Moreover, it is conceivable, if not likely, that the localization of TBC1D9B is subject to regulation, for example by altering nutrient supply as suggested by our findings in starved cells (see Fig. 3).

Based on the striking phenotypic similarity of *TBC1D9B* loss to the KO of *TMEM55B*, we hypothesize that the lysosomal membrane protein TMEM55B serves as a recruitment factor and/ or as an activator of

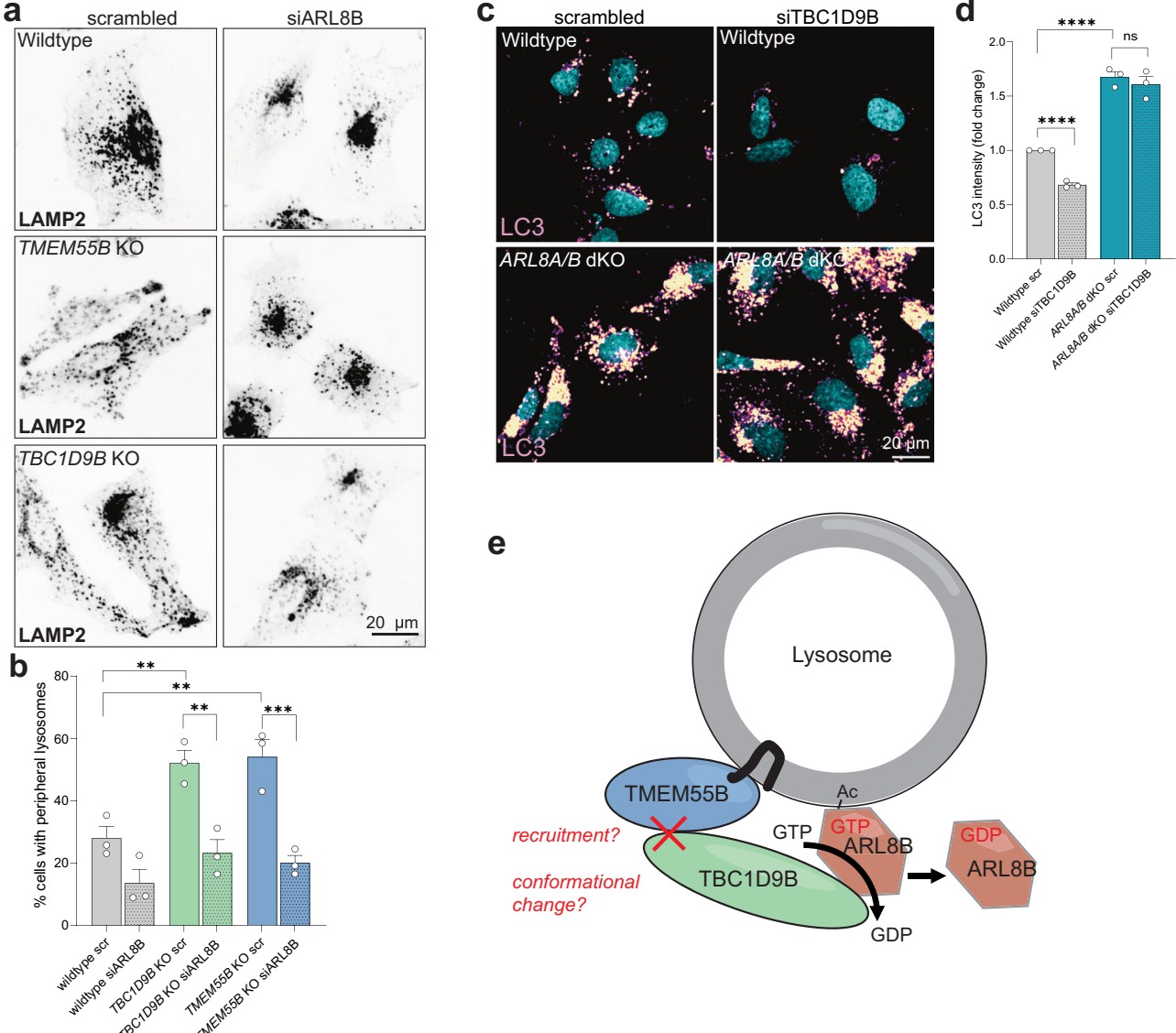

**Fig. 7 | TBC1D9B-mediated lysosomal positioning and autophagy defects are ARL8B-dependent. a** Confocal images of wildtype, *TMEM55B*-KO, and *TBC1D9B*-KO HeLa cells treated with scrambled siRNA or siRNA against ARL8B and stained for endogenous LAMP2. **b** The fraction of cells with peripheral lysosomes was quantified using OrgaMapper. One-way ANOVA (Šídák's multiple comparisons test), **$p < 0.01$, ***$p < 0.001$. wildtype scr vs. wildtype siARL8B $p = 0.1093$; wildtype scr vs. *TBC1D9B* KO $p = 0.0047$; scr wildtype scr vs. *TMEM55B* KO $p = 0.0026$; scr *TBC1D9B* KO scr vs. *TBC1D9B* KO $p = 0.0031$; siARL8B *TMEM55B* KO scr vs. TMEM55B KO siARL8B $p = 0.0003$; mean ± SEM from $n = 3$ independent experiments. **c** Confocal images of wildtype and *ARL8A/B* double-KO (DKO) cells treated with scrambled siRNA or siRNA against TBC1D9B and stained for LC3. **d** LC3-signal intensity was quantified and expressed as a fold change of scrambled-treated wildtype cells. One-way ANOVA, ns = not significant, ****$p < 0.0001$. Mean ± SEM from $n = 3$ independent experiments. **e** Schematic representation of the proposed model: TMEM55B binds TBC1D9B, which in turn acts as a GAP for ARL8B to facilitate nucleotide exchange and, thereby, membrane association and effector recruitment. TMEM55B-association putatively recruits TBC1D9B to the lysosomal surface and/or mediates a conformational change to activate TBC1D9B.

TBC1D9B activity. In preliminary experiments in HeLa cells overexpressing TMEM55B, we failed to detect any overt effects on TBC1D9B localization (Supplementary Fig. 6a). These negative results may be due to limitations imposed by protein overexpression or reflect TMEM55B's primary role in regulating TBC1D9B function, for example, by conformationally activating TBC1D9B to facilitate its catalytic activity. The association of TBC1D9B with other proteins and/or lipids, e.g., in response to physiological challenges, such as nutrient stress, may also steer its activity to distinct subcellular sites, e.g., to lysosomes (this study), starvation-induced vesicles[36], or recycling endosomes[38,52,53] via mechanisms that remain to be explored. Conversely, we do not yet understand how complex formation between TMEM55B and its various interactors, particularly its association with the dynein/kinesin adaptor JIP4[30,31] is regulated, e.g., by phosphorylation[54]. For example, whether TMEM55B can bind to JIP4 and TBC1D9B simultaneously or whether both proteins compete for TMEM55B remained unclear from our study. During the revision of this manuscript, a crystal structure of TMEM55B was published, and TMEM55B was identified as a central hub for adaptor recruitment on lysosomes, binding various clients via a TMEM55B-binding motif (TBM)[34,55]. Notably, a high number of TMM55B-interactors found in our study was also identified in that study (including JIP4, TBC1D9B, RNF213, BIRC6)[34]. These data suggest that different interactors compete for binding by a hitherto unknown regulatory mechanism[55]. Preliminary experiments suggest that TBC1D9B is dispensable for the reported perinuclear clustering phenotype induced by TMEM55B

overexpression (Supplementary Fig. 6a–c), which instead may reflect its association with JIP4 or other interactors, as reported earlier, supporting competitive binding of different clients[31,34]. Future studies will be needed to explore the underlying mechanisms in detail.

Several other open questions emanate from our study. For example, it is surprising that TBC1D9B appears to selectively act as a negative regulator of ARL8B function but not of the closely related ARL8A isoform, in spite of the profound sequence similarity (identity of both isoproteins (>90% identity)[18]. Similar findings have been reported for H-Ras, K-Ras, and N-Ras isoforms, which share over 90% sequence identity in their G-domains yet exhibit distinct biochemical properties, including differential interactions with effectors like Raf-RBD and for Rap1A and Rap1B proteins that are regulated by specific GAPs like RapGAP, with profound differences in effector engagement due to hypervariable regions and conformational preferences[56–58]. A prediction from our work, therefore, is that other GAP proteins must exist that act to physiologically control ARL8 activity in cells and tissues. Such GAP proteins could conceivably act selectively on ARL8A or both ARL8 isoforms, or exhibit cell- and tissue-specificity. Candidates are DENND6A, for which GAP activity was demonstrated in vitro[59], or other family members of the TBC family, including the TBC1D9B paralogs TBC1D9, TBC1D8, and TBC1D8B. Our experiments in *TBC1D9/TBC1D9B* double KO cells suggest that these proteins share similar functions. Adding to the overall complexity, TMEM55B has a close paralog: TMEM55A. TMEM55A does not bind JIP4[31], suggesting some degree of specificity, and its contribution to the TBC1D9B/ARL8B axis remains to be determined. Consistently, we observe that loss of TBC1D9B, i.e., a genetic condition resembling ARL8B gain-of-function, causes a much less severe lysosomal phenotype than the combined knockout of ARL8A and ARL8B (e.g., in human neurons as reported earlier by us;[20]) or the neurodegenerative phenotype reported for knockout mice lacking the myrlysin subunit of the ARL8A/ARL8B-activating BORC complex[51].

An equally important unanswered question pertains to the mechanism that integrates TBC1D9B into the cellular network of ARL8B effectors and their regulators, e.g., PLEKHM2/SKIP[10], KIF1A[19,20], HOPS[21], or RUFY1 or RUFY3 and 4[11,24] to exhibit contextual specificity of protein interactions at lysosomes and, possibly, other sites of ARL8B activity, such as recycling endosomes[23].

## Methods

### Ethics statement

Experiments involving human iPSC-derived neurons (iNs) were performed using the iPSC line BIHi005-A. BIHi005-A cells were generated by Sendai virus–mediated reprogramming of human dermal fibroblasts. Subsequently, the transcription factor NGN2 (originally provided by laboratories at Stanford University, CA, USA) was introduced under the control of a doxycycline-inducible rtTA element via CRISPR/Cas9-mediated genome engineering into the AAVS1 safe harbor locus. The donor material was fully anonymized and cannot be traced back to the individual. The cell line is registered in hPSCreg, and registration details, including IRB approval and signed informed consent information, are available at [https://hpscreg.eu/cell-line/BIHi005-A].

### Cell lines, antibodies, and reagents

**Chemicals and antibodies.** If not stated otherwise, analytical-grade chemicals were purchased from Sigma-Aldrich (MO, USA). The following primary antibodies were used:

| Reagent | Source | Identifier | RRID |
|---|---|---|---|
| Antibodies | | | |
| Actin | Sigma | Cat# A5441 | RRID:AB_476744 |
| AlexaFluor 488 anti-rat | Thermo/molecular probes | Cat# A11006 | RRID:AB_141373 |
| | | Cat# A11037 | RRID:AB_2534095 |
| AlexaFluor 594 anti-rabbit | Thermo/molecular probes | | |
| AlexaFluor 488 anti-mouse | Thermo/molecular probes | Cat# A21202 | RRID:AB_141607 |
| AlexaFluor 647 anti-mouse | Thermo/molecular probes | Cat# A21236 | RRID:AB_2535805 |
| AlexaFluor 568 anti-rabbit | Thermo/molecular probes | Cat# A11036 | RRID:AB_10563566 |
| AlexaFluor 647 anti-rabbit | Invitrogen | Cat# A21244 | RRID:AB_2535812 |
| ARL8A/B | Abcam | Cat# ab281997 | |
| ARL8B | Proteintech | Cat# 13049-1-AP | RRID:AB_2059000 |
| Calnexin | Proteintech | Cat#10427-2-AP | RRID:AB_2069033 |
| CF647 anti-mouse | Biotium | Cat# 20281 | RRID:AB_10853311 |
| Cox IV (clone 3E11) | Cell signaling | Cat# 4850 | RRID:AB_2085424 |
| EEA1 (clone C45B10) | Cell signaling | Cat# 3288 | RRID:AB_2096811 |
| GFP (clone 7.1 + 13.1) | Sigma Aldrich | Cat#11814460001 | RRID:AB_390913 |
| HA (clone 3F10) | Sigma Aldrich | Cat# 11867423001 | RRID:AB_390918 |
| Halo | Promega | Cat# G9211 | RRID:AB_2688011 |
| IRDye 800CW anti-mouse | LI-COR | Cat# 926-32210 | RRID:AB_621842 |
| IRDye 800CW anti-rabbit | LI-COR | Cat# 926-32211 | RRID:AB_621843 |
| LAMP1 | Cell signaling | Cat# 9091 | RRID:AB_2687579 |
| LAMP2 | Abcam | Cat# ab18528 | RRID:AB_775981 |
| LAMP2 (clone H4B4) | DSHB | Cat# sc-18822 | RRID:AB_626858 |
| LC3 | MBL | Cat# M152-3 | RRID:AB_10949134 |
| TBC1D9B | Bethyl laboratories | Cat# A304-655A | RRID:AB_2620850 |
| TMEM55B | Proteintech | Cat# 23992-1-AP | RRID:AB_2879391 |
| RAB8 | Cell signaling | Cat# 6975 | RRID:AB_10827742 |
| RAB11A | Invitrogen | Cat# 71-5300 | RRID:AB_87868 |

| Chemicals and reagents | | |
|---|---|---|
| Cell tracker green CMFDA | Invitrogen | Cat# C7025 |
| Complete, mini protease inhibitor cocktail | Roche | Cat# 11836153001 |
| Digitonin | Invitrogen | Cat# BN2006 |
| FastDigest BpiI | Thermo | Cat# FD1014 |
| GFP-Trap® magnetic agarose | Proteintech | Cat# gtma |
| Goat serum | Gibco | Cat# 16210072 |
| Igepal | Sigma-Aldrich | Cat# CA-630 |
| Intercept (TBS) blocking buffer | LI-COR | Cat# 927-60001 |
| INTERFERin | VWR | Cat# 101000028 |
| jetPrime | VWR | Cat# 101000001 |
| Matrigel | Corning | Cat# 356231 |
| NucleoSpin tissue | Macherey-Nagel | Cat# 740952.250 |
| Phosphatase inhibitor cocktail 2 | Sigma-Aldrich | Cat# P5726 |
| Phosphatase inhibitor cocktail 3 | Sigma-Aldrich | Cat# P0044 |

| | | | |
|---|---|---|---|
| Puromycin | Gibco | Cat# A1113803 | |
| Saponin | Sigma | Cat# 47036 | |
| Shandon Immu-Mount | Epredia | Cat# 9990402 | |
| Oligonucleotides (siRNA) | | | |
| siARL8B | Dharmacon/Horizon | Cat# L-031872-02 | |
| Sigma scr control | Sigma Mission® siRNA universal negative control #1 | Cat# SIC001 | |
| siTBC1D9B | Dharmacon/Horizon | Cat# L-020294-01 | |

## Cell lines

U2OS and HeLa cells were sourced from ATCC. The cell lines were cultured in DMEM medium containing 4.5 g/ml glucose and L-glutamine (Gibco) supplemented with 10% heat-inactivated fetal calf serum (Gibco) and 100 U/ml penicillin and 100 µg/ml streptomycin (Gibco). Cells were cultured in Dulbecco´s modified Eagle medium (DMEM, Gibco) with 10% (v/v) fetal calf serum (FCS, Thermo Fisher) and Penicillin/Streptomycin (PenStrep) at 37 °C with a 5% $CO_2$ atmosphere. Cell lines were routinely tested for Mycoplasma contamination, and all tests were negative.

## Plasmids and cloning

hTBC1D9B-mCherry was a kind gift Drs. Xiao-Ming Yin and Yong Liao (Indiana University-Purdue University Indianapolis) and subcloned into the pcDNA3.1 Hygro (+) vector with a C-terminal HA-tag with HindIII and BamHI and into the pEGFP-N1 vector with a C-terminal GFP-tag. The cDNA encoding full-length TBC1D9B WT-GFP and TBC1D9B WT was subcloned from the mammalian expression vector (peGFP-N1-TBC1D9B) into pFL-His10-StrepII vector through Gibson assembly. Additionally, the GAP mutant region was subcloned from the mammalian expression vector (pcDNA3.1 Hygro(+)-TBC1D9B GAP mutant) into pFL-10His-StrepII-TBC1D9B WT vector through the KasI and SbfI sites within TBC1D9B.

| Plasmids | Backbone | Insert gene | Insert species |
|---|---|---|---|
| ARL8B-WT-GFP | pEGFP-N1 | ARL8B with GFP tag | Human |
| ARL8B-QL-GFP | pEGFP-N1 | ARL8B Q75L with GFP tag | Human |
| ARL8B-TN-GFP | pEGFP-N1 | ARL8B T34N with GFP tag | Human |
| GST-ARL8B | pGEX6P1 | ARL8B with GST tag | Human |
| GST-RAB11A | pGEX-5X | RAB11A with GST tag | Canine |
| TMEM55B-HA | pcDNA3.1 Hygro (+) | TMEM55B with HA tag | Mouse |
| TMEM55B-GFP | pEGFP-N1 | TMEM55B with GFP tag | Mouse |
| TMEM55B-Δ13-65-HA | pcDNA3.1 Hygro (+) | TMEM55B with deletion of amino acids 13–65 with HA tag | Mouse |
| TBC1D9B-HA | pcDNA3.1 Hygro (+) | TBC1D9B with HA tag | Human |
| TBC1D9B-GFP | pEGFP-N1 | TBC1D9B with GFP tag | Human |
| | | TBC1D9B$^{RYQ/AAA}$ with HA tag | Human |
| TBC1D9B$^{RYQ/AAA}$-HA | pcDNA3.1 Hygro (+) | | |
| TBC1D9B-TBC-HA | pcDNA3.1 Hygro (+) | TBC domain of TBC1D9B with HA tag | Human |
| TBC1D9B-CT-HA | pcDNA3.1 Hygro (+) | C-terminal domain of TBC1D9B with HA tag | Human |
| TBC1D9B-NT-HA | pcDNA3.1 Hygro (+) | N-terminal domain of TBC1D9B with HA tag | Human |
| TBC1D9B-TBC+CT-HA | pcDNA3.1 Hygro (+) | TBC and C-terminal domain of TBC1D9B with HA tag | Human |
| TBC1D9B-TBC+NT-HA | pcDNA3.1 Hygro (+) | TBC and N-terminal domain of TBC1D9B with HA tag | Human |

## Mutagenesis of TBC1D9B

Mutations were inserted using mutagenesis or fusion-PCR. PCR mixtures were in both cases prepared with 0.2 mM dNTPs, 1x Phusion GC buffer, 0.2 µM of overlapping primers containing the desired mutation, 3% DMSO, and Phusion high-fidelity DNA polymerase. The PCR reaction was carried out in a FlexCycler 2 PCR device (Analytik Jena) with 98 °C denaturation, 45–55 °C annealing, and 72 °C elongation temperature. For the site-directed mutagenesis, the PCR product was incubated with DpnI at 37 °C for 1 h to digest the PCR template before transformation. Two sets of PCR were needed for the fusion-PCR. A first set to create at the mutation ending and beginning PCR products with an overlap, and another PCR to fuse these two PCR products.

## CRISPR/Cas9-mediated gene editing

*TMEM55B*, *TBC1D9*, and *TBC1D9B* CRISPR/Cas9 mediated HeLa knockout cells were generated with CRISPR guide RNAs ("gene knockout kit") from Synthego (Redwood City, CA, USA) with the following sequence: *TBC1D9:* CGGCGGCGGACUGGCGGGUA, ACCCAUACUUCAUCCUGCAG, CAUGUGGUGAACCCGGAGG, *TBC1D9B*: AGGGUGCCCACGAGAAGACC, CCUGUUCCCAGGUCUUCUCG. For the *TMEM55B* gene knockout, guides from the Synthego "gene knockout kit" with the following sequence were used: ACCGGCUCCGUACGGUGGUG, ACGGUUUAGUGGGGCCCGGC, GUGGCGGCCAUGGCGGCAGA. Genomic HA-tagged TBC1D9B HeLa Cells were generated with CRISPR/Cas9 mediated knockins using Alt-R sgRNA and ssODN Donors from IDT (Coralville, IA, USA) with the following sequences: Alt-R sgRNA: AGGCATCAGCCGGAAACTCC ssODN Donor: TGATGACACCTTGGGCCAGGCCTCACAGCTGCAGGCATCAGGCGTAGTCGGGGACGTCGTAGGGGTAGCCGGAAACTCCAGGCTGCTCATGGTCACTGGCGGTGCTGAACTGTCTC. CRISPR/Cas9-mediated knockout and knockin was performed using the Neon™ Transfection System (Thermo Fisher Scientific): Synthetic CRISPR guide RNAs, ssODN Donors and recombinant Cas9 (IDT) were electroporated according to the manufacturer's protocol and recommendations (1005 V, 35 ms, 2 pulses). The medium was exchanged after reaching 90% confluence the next day, and diluted cells were seeded on 96-well plates to obtain single-cell clones. The single clones were expanded, and the editing was validated by immunoblot with specific antibodies against TBC1D9B and TMEM55B. The knock-in was additionally validated by Sanger sequencing and Synthego analysis tool (https://design.synthego.com/#/validate) analysis with the following Primers: amplification: CCGACTGGTGCATCTCCTTT and GAGAGCCTGTAATGCTAGGCAG sequencing: GAGAGCCTGTAATGCTAGGCAG.

*U2OS ARL8 double knockout (DKO) cell line generation*: pSpCas9(BB)-2A-Puro (PX459) V2.0 was sourced from Addgene (#62988). Oligonucleotides with gRNA sequences were annealed and

cloned into the PX459 plasmid using the FastDigest BpiI (Thermo) restriction site. The following gRNA target sequences were employed to generate *ARL8* DKO cell line: sgRNA for *ARL8A*: CGCGATCACGTTG ACGAAGG, and sgRNA for *ARL8B*: AGAGATGGAGCTGACGCTCG[20].

The PX459 plasmids containing the corresponding sgRNAs were co-transfected into U2OS cells using jetPRIME, following the manufacturer's instructions. Co-transfection was repeated after 48 h. Two days after the second transfection, puromycin selection (2 μg/ml) was applied to eliminate cells that had not been transfected with CRISPR plasmids. The selection process was maintained for 10 days, with medium changes every 2 days to replenish puromycin.

To generate clonal cell lines, limiting dilution cloning was conducted by plating cells into a 96-well plate at a density of 0.5–1 cell per well. Colonies derived from single cells were allowed to expand. Genomic DNA was then isolated with the help of NucleoSpin Tissue XS from the clonal cell lines and subjected to sequencing to identify frameshift mutations by amplifying the genomic regions with the following primers: for *ARL8A* KO, forward 5'- GATTCCGGGGCAG CGAGTCGT-3' and reverse 5'-TACTAGGCCCCAAGCGCTCG-3', and for *ARL8B* KO, forward 5'-TCATCTCCCGCCTGCTGGACT-3' and reverse 5'-TCCAAATGCTGGCCCTCGG-3'. KO efficiency was assessed based on Sanger sequencing data using the Synthego analysis tool (https://design.synthego.com/#/validate). Additionally, DKO clones were verified by Western blot. Correct cell clones were expanded and cryopreserved in liquid nitrogen.

### Transfection of cells with plasmid DNA
Cells were transfected using the proprietary cationic polymer Turbo-FectTM (Thermo Fisher). For transfection 1 μL TurboFectTM per 500 μg DNA was used and incubated with the DNA in serum-free media for 20 min at room temperature. The mixture was dripped on adherent cells in DMEM with 10% FCS. The medium was changed after 4 h.

### siRNA-transfection
For *ARL8B* knockdown, cells were seeded into 6-well plates at a density of $2.5 \times 10^5$ cells per well and subjected to reverse transfection with 50 nM siRNA using INTERFERin as per the manufacturer's protocol. This transfection was repeated 24 h later. For *TBC1D9B* KD, reverse transfection was similarly performed, with a subsequent repeat after 24 h, utilizing 100 nM siRNA and jetPRIME, following the manufacturer's instructions. Cells were harvested and analyzed 72 h post-initial transfection.

### Starvation
Cells were washed three times with EBSS (Gibco) and incubated in EBSS for 6 h to induce starvation.

### Preparation of cell lysates and immunoblotting
To verify the knockdown of *TMEM55B* and *TBC1D9B*, cells were lysed on ice in PBS with 1%(v/v) Triton X-100 and 1x cOmplete Protease inhibitor cocktail (Roche) for 60 min, followed by 3x 12 s sonication with a Branson Sonifier 450 (Emerson Industrial Automation). Cell lysates were centrifuged at $12,000 \times g$ for 15 min at 4 °C. Protein concentration was measured with the Pierce™ BCA Protein Assay Kit (Thermo Fisher Scientific). For the SDS page of cell lysates and immunoprecipitation, the samples were denatured with Laemmli buffer at 95 °C for 10 min. Samples were separated via SDS-PAGE using 10% or 12.5% acrylamide gels and, in a semi-dry western blot, transferred to either nitrocellulose (GE Healthcare) or PVDF (GE Healthcare) membranes. Detection was carried out with the Lumigen ECL Ultra Kit (Beckman Coulter) in an Amersham Imager 680 (GE Healthcare).

To verify the knockdown of ARL8B and TBC1D9B, cells were lysed in a buffer containing 1% Igepal, 1 M Tris pH 8, 150 mM NaCl, and protease inhibitors cComplete mini, along with phosphatase inhibitor cocktails 2 and 3. Cell lysates were clarified by centrifugation, and the resulting supernatant was denatured with Laemmli buffer at 95 °C for 10 min. Lysates were resolved in 4–20% SDS-PAGE gels and transferred onto nitrocellulose membranes using transfer buffer (20% methanol). Membranes were blocked for 45 min at RT in Intercept TBS Blocking Buffer, and further incubated with primary antibodies diluted in the same blocking buffer (overnight, at 4 °C). After overnight incubation, membranes were washed 3 times with TBST, incubated with IRDye secondary antibodies (1 h at RT), washed again 3 times again with TBST, and visualized using the LI-COR Odyssey FC 2800 imager. Quantification was performed using Fiji. Uncropped blots are presented in the Supplementary Information.

### Proximity labeling proteomic analysis of ARL8B-TurboID-expressing iPSC-derived human neurons
Human iPSCs were differentiated into cortical neurons using an optimized method based on the rapid neuronal differentiation protocol described in refs. 20,60. WT iPSC-derived neurons were transduced with lentivirus expressing ARL8B-TurboID and TurboID-NES, respectively. A negative control group is also needed, in which no TurboID lentivirus was transduced. iNeurons were incubated with 50 μM biotin for 8 h to initiate labeling on DIV 15. Then stop the labeling by moving cells onto ice and washing five times with DPBS. Neurons were lysed in RIPA lysis buffer, and biotinylated protein was enriched with streptavidin beads (Thermo Fisher Scientific, cat. no. 88817). Elute the enriched protein from the beads by boiling the sample in Laemmli sample buffer with 2 mM biotin and 20 mM DTT at 95 °C for 10 min, and resolved by SDS-PAGE. Two bands per lane were cut by hand, and in-gel tryptic digestion was performed. Tryptic peptides were analyzed by a reversed-phase capillary liquid chromatography system (Ultimate 3000 nanoLC system; Thermo Scientific) connected to an Orbitrap Fusion mass spectrometer and Orbitrap Fusion Lumos mass spectrometer (Thermo Scientific). Identification of proteins was performed using MaxQuant (version 1.6.26) software. Data were searched against the UniProt human protein database. The initial maximum mass deviation of the precursor ions was set at 20 ppm, and the maximum mass deviation of the fragment ions was set at 0.35 Da. Methionine oxidation was used as the variable modification, Cys carbamidomethyl was used as the fix modification. False discovery rates were <1% based on matches to reversed sequences in the concatenated target-decoy database. Proteins were considered if at least two sequenced peptides were identified (at least one unique peptide and at least two razor + unique peptides).

### Affinity purification mass spectrometry (AP-MS)
For GFP-immunoprecipitation, HEK293 cells were transfected using x-tremeGeneTM (Roche, Rotkreuz, Switzerland) with eGFP-TMEM55B or GFP as a negative control. After 2 days of expression, cells were harvested in ice-cold PBS and subsequently lysed in AP-MS buffer (10 mM Tris/Cl pH 7.5, 150 mM NaCl, 10% Glycerol, 0.5 mM EDTA, 0.5% Nonidet™ P40) supplemented with protease inhibitor cocktail (Roche, Rotkreuz, Switzerland) by sonication, followed by centrifugation ($20,000 \times g$ for 10 min. at 4 °C) to remove cellular debris. Supernatants were subjected to immunoprecipitation using GFP-Trap beads (Proteintech, Planegg-Martinsried, Germany) blocked with 5% BSA in AP-MS buffer for 1 h at 4 °C. Beads were collected by centrifugation at 4 °C and washed three times with AP-MS buffer and two times with AP-MS buffer without detergent. After washing, on-bead trypsin digestion was performed, following the GFP-Trap protocol, using sequencing-grade modified trypsin (Promega, Walldorf, Germany); see below.

### Sample preparation for AP-MS
Protein pellets were resuspended in 50 μL of 8 M urea dissolved in 50 mM ammonium bicarbonate (ABC) buffer. The same volume of 0.2% ProteaseMAX (Promega, Cat# V2072) solution in ABC buffer was added and incubated for one hour with vortexing. The disulfide bonds in

proteins were reduced with 5 mM Tris(2-carboxyethyl)phosphine (TCEP) for 20 min at RT, followed by alkylation with 10 mM iodoacetamide (IAA). Tubes were incubated in the dark for 15 min and immediately quenched with excess (25 mM) of TCEP prepared in ABC. Subsequently, proteins were digested overnight at 37 °C using MS-grade trypsin (Promega, Cat# V5280). The next morning, the digestion reaction was stopped by acidification using 1% formic acid (FA). Desalting using C18 spin columns (Thermo Scientific, Cat# 89,870) was performed according to the manufacturer's instructions. Peptide solutions were dried down in a refrigerated speed vac and stored at −80 °C.

### Tandem mass spectrometry analysis of affinity-purified samples
Three micrograms of each fraction or sample were auto-sampler loaded with an UltiMate 3000 HPLC pump onto a vented Acclaim Pepmap 100, 75 µm × 2 cm, nanoViper trap column (Thermo Fisher Scientific, Cat# 164535) coupled to a nanoViper analytical column (Thermo Fisher Scientific, Cat# 164570, 3 µm, 100 Å, C18, 0.075 mm, 500 mm) with stainless steel emitter tip assembled on the Nanospray Flex Ion Source with a spray voltage of 2000 V. An Orbitrap Fusion (Thermo Fisher Scientific) was used to acquire all the MS spectral data. Buffer A contained 94.785% $H_2O$ with 5% ACN and 0.125% FA, and buffer B contained 99.875% ACN with 0.125% FA. Each sample was analyzed in a single-shot analysis lasting 150 min.

We used a CID-$MS^2$ method. Briefly, ion transfer tube temp = 300 °C, Easy-IC internal mass calibration, default charge state = 2, and cycle time = 3 s. Detector type set to Orbitrap, with 60 K resolution, with wide quad isolation, mass range = normal, scan range = 300–1500 m/z, max injection time = 50 ms, AGC target = 200,000, microscans = 1, S-lens RF level = 60, without source fragmentation, and datatype = positive and centroid. MIPS was set as on, included charge states = 2–6 (reject unassigned). Dynamic exclusion enabled with $n = 1$ for 30 s and 45 s exclusion duration at 10 ppm for high and low. Precursor selection decision = most intense, top 20, isolation window = 1.6, scan range = auto normal, first mass = 110, collision energy 30%, CID, Detector type = ion trap, OT resolution = 30 K, IT scan rate = rapid, max injection time = 75 ms, AGC target = 10,000, $Q = 0.25$, inject ions for all available parallelizable time.

### AP-MS data analysis and quantification
Protein identification/quantification and analysis were performed with Integrated Proteomics Pipeline-IP2 (Bruker, Madison, WI) using ProLuCID,[61,62] and DTASelect2[33,63]. Spectrum raw files were extracted into MS1, MS2, and MS3 (For TMT experiments) files using RawConverter (http://www.scripps.edu/yates/Software.html). The tandem mass spectra were searched against the UniProt human protein database, including isoforms (downloaded on 01-01-2014), and matched to sequences using the ProLuCID/SEQUEST algorithm (ProLuCID version 3.1) with 5 ppm peptide mass tolerance for precursor ions and 600 ppm for fragment ions. The search space included all fully and half-tryptic peptide candidates within the mass tolerance window with no-mis cleavage constraint, assembled, and filtered with DTASelect2 through IP2. To estimate peptide probabilities and false-discovery rates (FDR) accurately, we used a target/decoy database containing the reversed sequences of all the proteins appended to the target database (UniProt). Each protein identified was required to have a minimum of one peptide of minimal length of six amino acid residues within 10 PPM of the expected m/z; however, this peptide had to be an excellent match. After the peptide/spectrum matches were filtered, we estimated that the protein FDRs were ≤1% for each sample analysis. Resulting protein lists include subset proteins to allow for consideration of all possible protein forms implicated by at least two given peptides identified from the complex protein mixtures.

### Protein expression and purification
His$_{10}$-tagged TBC1D9B wildtype and TBC1D9B$^{RYQ/AAA}$ GAP mutant proteins were expressed in Sf21 insect cells using SF900-II serum-free medium (Thermo Fisher Scientific). Briefly, Sf21 cells (800 ml) were grown to a density of approximately $1.5 \times 10^6$ cells per ml and infected with 8 ml of amplified baculovirus. Cells were harvested 48 h post-infection. For purification, cell pellets were resuspended in 35 ml of lysis buffer (50 mM Tris, pH 7.5, 300 mM NaCl, 10 mM imidazole, 1 mM dithiothreitol (DTT), 0.5% Triton X-100, and one tablet of protease inhibitor cocktail per 50 ml), sonicated for 1 min (1 s on, 5 s off), and centrifuged at $87,000 \times g$ for 20 min. The supernatant was collected and incubated with 0.5 ml of Ni-NTA agarose beads (Sigma) on a rotating wheel for 1 h at 4 °C. The beads were washed once with 10 ml of lysis buffer, followed by three washes with 10 ml of wash buffer (50 mM Tris, pH 7.5, 300 mM NaCl, 20 mM imidazole, 1 mM DTT). Proteins were eluted with 4 ml of elution buffer (20 mM Tris, pH 7.5, 300 mM NaCl, 300 mM imidazole, 5 mM DTT). The Ni-NTA eluate was further purified by size-exclusion chromatography (SEC) using a buffer containing 20 mM Tris, pH 7.5, 300 mM NaCl, and 5 mM DTT at 4 °C. Purified proteins were concentrated to approximately 0.4 mg/ml for TBC1D9B WT and 3 mg/ml for the GAP mutant. Proteins were either used immediately for biochemical assays or flash-frozen in liquid nitrogen and stored at −80 °C for future use.

For the expression of GST-ARL8B, the plasmid was transformed into *E.coli* BL21 DE3 cells and induced with 0.5 mM IPTG overnight. The cells were lysed in 20 mM Tris pH 7.5, 150 mM NaCl, 5 mM EDTA, 0.1% Triton-X, 5 mM DTT, and one protease inhibitor tablet (Roche) by ultrasonication. The lysate was centrifuged at 27,000 rpm for 20 min. The supernatant was incubated with pre-washed glutathione-coupled beads from Novagen on a rotation wheel for 1.5 h. For GST-ARL8B, the beads were washed with 20 mM Tris pH 7.5, 150 mM NaCl, 1 mM EDTA, and eluted with 50 mM reduced glutathione.

### GST-pulldown assay
The GST-pulldown assay was performed with immobilized GST, GST-fused ARL8B Q75L (a constitutively active mutant), and ARL8B T34N (a dominant-negative mutant). Fifteen micrograms of immobilized GST-fused ARL8B Q75L and ARL8B T34N were complexed with 0.2 mM GTPγS or GDP in 500 µl binding buffer (50 mM Tris pH 7.5, 150 mM NaCl, 5 mM DTT, 5 mM MgCl$_2$, 0.02% Triton X100) overnight at 4 °C. Binding assays were started by adding 10 µg purified His10-TBC1D9B-GFP. Samples were incubated on a rotation wheel for 1 h at 4 °C. Unbound material was washed out by 3x washes with binding buffer, and bound proteins were eluted with 2x SDS-PAGE sample buffer. The samples were analyzed by SDS-PAGE and followed by GFP in-gel fluorescence or Coomassie blue staining.

### In vitro GTPase-Glo assay
In vitro GTPase-Glo assay was performed using Promega's GTPase-Glo Assay kit (V7861). His$_{10}$-tagged TBC1D9B wildtype (WT) and TBC1D9B$^{RYQ/AAA}$ GAP mutant proteins were diluted in GAP buffer containing 5 µM GTP and 1 mM DTT in the absence or presence of GST-ARL8B or GST-RAB11A protein. Reaction mixes were incubated at 37 °C for the indicated times. Once the GTPase reaction was completed, 5 µl of reconstituted GTPase-Glo reagent was added to all the wells at room temperature. The plate was incubated at room temperature for 30 min with gentle shaking. At the end of the incubation period, 10 µl of the Detection reagent was added to all the wells. The luminescence signal was measured using a TECAN (Infinite M PLEX) plate reader after the plates were further incubated for 10 min at room temperature.

### Immunofluorescence microscopy
Cells were grown on glass coverslips and fixed with 4% (w/v) paraformaldehyde (PFA) in PBS for 20 min at room temperature and permeabilized with 0.2% (w/v) saponin in PBS for 5 min at room temperature. PFA-derived fluorescence was quenched with 0.12% (w/v) glycin in 0.2% saponin-PBS for 10 min at room temperature. Blocking was carried out in a 10% (v/v) FCS in 0.2% (w/v) saponin-PBS blocking

solution for 1 h at room temperature. Coverslips were incubated overnight at 4 °C with the primary antibody diluted in blocking solution and subsequently incubated at room temperature for 1 h with the fluorophore-coupled secondary antibody (Alexa Fluor 488, 594, and 647).

For quantitative imaging and automated image analysis, cells were seeded at a density of $5 \times 10^4$ on 12 mm coverslips coated with Matrigel in 24-well plates. The following day, the cells were treated with 10 μM CMFDA for 1 h, then fixed in 4% paraformaldehyde/4% sucrose in PBS (15 min at RT). After three washes with PBS, the cells were incubated with primary antibodies (1:100 in 10% goat serum/0.05% saponin in PBS for 1 h, at RT). The cells were then washed three times with PBS and incubated with fluorophore-conjugated secondary antibodies (1:500 in 10% goat serum/0.05% saponin/PBS, with DAPI for 1 h at RT). After the incubation, the cells were washed three times with PBS and mounted using Shandon Immu-Mount.

For LC3 staining, the permeabilization step was additionally included, performed with a buffer containing 20 μM digitonin (in 10% goat serum in PBS for 10 min, at RT). The cells were then washed three times with a buffer consisting of 10% goat serum and PBS. From this point onward, the detergent was excluded from the protocol, while all other steps were carried out as described above.

Finally, the coverslips were mounted on glass slides and embedded in Mowiol containing 1 mg/ml 4′, 6-diamidino-2-phenylindol (DAPI). Fixed cells were analyzed with a ZeissLSM 980 fluorescence microscope equipped with an automated stage and the ZEN 3.1 (blue edition) software (Carl Zeiss Microscopy GmbH) or with NikonCSU Spinning Disk using a 40x air objective (NA = 0.95), hardware autofocus, and semi-automatic multi-position image acquisition: Random fields of view were selected based on DAPI signal, saved in the software, and imaged all at once.

## Analysis of lysosome positioning
Lysosome positioning analysis in fixed cells was carried out using the OrgaMapper Fiji Plugin[39]. Briefly, the Plugin segments individual nuclei, filters them by size, excludes nuclei touching the edges of the field of view, segments single cells using background subtraction and seeded watershed segmentation, and detects lysosomes using the Laplacian of Gauss (LoG) filter and the Find Maxima command. The distance of each lysosome detection within the individual cell mask is then measured based on a Euclidean distance map from the edge of the nuclei mask. R scripts were used to compile all measured parameters into single Excel files for further analysis. The percentage of cells with peripheral lysosomes was calculated as previously[39] to cancel out possible biases of cell size on lysosome positioning measurements. Briefly, the average distance of lysosomes to the nucleus, as well as the average distance normalized by the cell's Feret's diameter, was computed for the entire cell population of each experiment. Cells with above-average distance as well as above-average distance normalized by the cell's Feret's diameter were regarded as cells with a true peripheral lysosome phenotype. The percentage of cells with true peripheral lysosomes in each sample was plotted.

## Lysosomal motility measurements
HeLa cells were grown on ibidi μ-Slide 8-well high dishes and grown to approximately 70% confluency. The cells were loaded with Lysotracker red, and imaged on a Zeiss LSM 980 Airyscan 2 microscope with a 63× oil objective equipped with environmental control (37 °C, and 5% CO₂). Images were recorded in a 0.1 s interval with 10 images in the Z-direction (0.5 μm per image) over 3 min. For starved conditions, HeLa wildtype and knockout cells were grown to near confluency in Greiner Bio-One 10-well glass-bottom dishes, starved in EBSS, loaded with Lysotracker red, and imaged live on a NikonCSU W1 spinning disc microscope equipped with environmental control (humidity, 37 °C, and 5% CO₂) using a 60x oil objective. Time-lapse sequences were

recorded with a 0.5-s interval for 15 s. Analysis was performed per field of view using the TrackMate Fiji plugin.

## DQ-BSA degradation assay
For degradation analysis, 100.000 HeLa cells were seeded on glass coverslips in a 12-well plate and incubated overnight (37 °C, 5% CO₂). The cells were subsequently incubated with 10 μg/ml DQ-BSA (Thermo Fisher Scientific) in either DMEM with 1% FCS, 1% NEAA (Gibco), and 1% 1 M HEPES (Gibco) or EBSS for 6 h. Microscopy samples were prepared as described previously. Corrected total cell fluorescence (CTCF) measurement was carried out in ImageJ.

## Dextran endocytosis assay
HeLa cells were prepared for equal to DQ-BSA degradation measurements. After overnight incubation, the cells were incubated for 30 min with 0.5 mg/ml dextran-Texas Red® (70.000 MW, Life Technologies) in DMEM supplemented with 1 mg/ml BSA, washed 3 times with PBS, and cultivated in DMEM with 10% FCS or EBSS for 6 h. The cells were fixed and imaged as described previously and analysed using ImageJ.

## LC3 and SiR-lysosome intensities
All measurements were performed in the OrgaMapper Fiji plugin[39]. Single cells were segmented, LC3 or SiR-lysosome punctae were detected, and mean intensities on the detections were calculated per cell.

## Measurement of p62 levels
Cells seeded on Matrigel-coated coverslips were washed 5 times in EBSS, then supplemented with full media or EBSS, and treated with bafilomycin A1 as indicated. Cells were incubated for 3 h, and then supplemented additionally with 1 μM CMFDA. Cells were finally washed once in PBS, fixed in 4% PFA/4% sucrose in PBS for 20 min, washed twice in PBS, and stored in PBS.

For immunostainings cells were blocked for 30 min in blocking buffer (10% GS, 0.3% Triton-X-100, 20 mM HEPES pH 7.4), then incubated with anti-p62 antibody (1:200, BD Bioscience, 610832), washed three times for 10 min in PBS, incubated with AF568-labeled goat-anti-mouse antibody, washed again three times in PBS for 10 min, and finally mounted using Shandon™ Immu-Mount™ containing Hoechst33342.

Imaging was performed on a Nikon TiE2 (Nikon) equipped with a confocal spinning-disk unit (CSU-W1, Yokogawa) and a PL APO λ 40x/ 0.95 NA air objective, without additional magnification. The microscope was equipped with two sCMOS cameras (pco.edge, 4.2bi, 6.5 μm/pixel, 2048 × 2048 pixel) and controlled with NIS-Elements software (Nikon). Images were taken with the following sequential fluorophore settings: Hoechst33342 (Ex.: 405 nm; Em.: 420–460 nm), CMFDA (Ex.: 488 nm; Em.: 500–550 nm), AF568 (Ex.: 561 nm; Em.: 574–626).

Image processing and quantitative analysis were performed with the open-source software Fiji (ImageJ). Automated single-cell segmentation, p62 detection, and intensity measurements were carried out using the custom-made ImageJ plugin OrgaMapper[39].

All data are presented as mean ± SEM, and were obtained from ≥ 3 independent experiments with total sample numbers provided in the figure legends. Statistical significance was evaluated using Student's $t$ test or one-way ANOVA and Dunnett´s multiple comparison test. Significant differences were marked as $*p < 0.05$, $**p < 0.01$, $***p < 0.001$, $****p < 0.0001$.

## Measurement of autophagic flux using Halo_LC3 stable cell lines
Cells were seeded into six-well dishes and treated as indicated. Media were supplemented with 100 nM Halo ligand CA-JFX650. Cells were washed once in PBS, harvested by trypsinization, washed once in PBS, and resuspended in 75 μl RIPA buffer containing protease inhibitors.

Upon 15 min incubation on ice, cell lysates were cleared by centrifugation for 15 min at $10,000 \times g$ and subjected to SDS-PAGE and Western blot analysis. For each experiment, samples were loaded as technical duplicates. The incorporated Halo-ligand was detected by in-gel fluorescence on a Biorad ChemiDoc Imaging System controlled by Image Lab software, version 6.1.0. Halo-tag was detected by quantitative Western blotting using anti-Halo primary antibody (Promega, #G9211, 1:1000) and a fluorescently labeled secondary antibody. The fluorescent signal was detected on a LICORbio detection system controlled by ImageStudioLite software.

### Data and reproducibility

All statistical analyses were performed using GraphPad Prism version 7.04 or later. The statistical tests used are indicated in the figure legend. It is also indicated whether the error bar represents the SD or SEM. No sample-size calculation was performed, but statistical methods were used to calculate standard deviation as noted in the figure legends. Experiments were performed with at least $n = 3$; $n$-numbers are provided in the figures/figure legends. Data were only excluded if obvious technical problems occurred during the experiments. Generally, no data were excluded. Unless otherwise stated, experiments with representative results were repeated at least three times. The number of independent experiments is provided in the figure legends. Statistical significance is represented with the following symbols: $*p \leq 0.05$; $** \leq 0.01$; $***p \leq 0.001$; $****p \leq 0.0001$; ns = no statistical difference. For all treatments, cultured cells of the required genotype/treatment were randomly allocated to the control and genotype/treatment groups.

### Reporting summary

Further information on research design is available in the Nature Portfolio Reporting Summary linked to this article.

## Data availability

All data underlying the findings presented in this paper are either contained within the paper or its Supplementary files. In addition, Source data are provided within the source data file. The raw and processed mass spectrometry data have been deposited in the MassIVE and ProteomeXchange repositories (identifiers MSV000095685 [https://massive.ucsd.edu/ProteoSAFe/dataset.jsp?task=f549fe6470184dd59a858d024e2a428c] and PXD055148. Source data are provided with this paper.

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

## Acknowledgements

We thank Sebastian Held (CAU, Kiel), Delia Löwe (FMP, Berlin), Silke Zillmann (FMP, Berlin), and Claudia Bahnik (FMP, Berlin) for their excellent technical support. We are grateful to Prof. Noboru Mizushima (University of Tokyo, Japan) for the kind gift of the stable Halo-LC3 HeLa cell line. We are also indebted to Prof. Oliver Daumke and Dr. Marius Weismehl (MDC, Berlin) for help with GAP assays and critical discussions, the FMP mass spectrometry core facility, especially to Heike Stephanowitz and Prof. Fan Liu for proteomic analysis. Prof. Mahak Sharma is acknowledged for the gift of RAB11A constructs. The study was supported by the Deutsche Forschungsgemeinschaft (DFG) (FOR2625) to M.D. (DA 1785/2-2), V.H. (TRR186/A08; HA2686/26-1; Germany's Excellence Strategy—EXC-2049–390688087), and M.S. (SCHW866/6-1 and 7-1). J.N.S. received funding from the NIH (R01AG078796 and S10OD032464). W.T.L. acknowledges the Yushan Fellow Program by the Ministry of Education (MOE), Taiwan, for the financial support (MOE-113-YSFAG-0006-002-P1). V.H. further acknowledges support by the European Commission (ERC Advanced Grant 884281 "SynapseBuild"). Confocal microscopy (CAU Kiel) was supported by the DFG (INST257/640-1FUGG).

## Author contributions

M.E., K.K. are equally contributing second authors. V.D. aided by H.L.S., M.R. conducted cell biological and biochemical experiments related to TMEM55B and its interaction with TBC1D9B, M.T. conducted ARL8B GAP and interaction studies as well as cell biological experiments, W.T.L. expressed and purified TBC1D9B protein and performed the GST pull-down experiment, M.V. helped in generating TBC1D9B-knockin cells, and V.D., M.K., K.K., and M.E. analyzed lysosome distribution and conducted functional cell biological experiments, M.S. performed GFP-Nanotrap pulldown experiments for AP-MS, and J.N.S. performed mass spectrometry of GFP-Nanotrap pulldown experiments, M.D., V.H.

conceived and supervised the overall study, and wrote the paper with input from all authors. Correspondence and requests for materials should be addressed to M.D. (markus.damme@uni-bielefeld.de) or V.H. (haucke@fmp-berlin.de).

## Funding

## Competing interests

The authors declare no competing interests.

## Additional information

**Supplementary information** The online version contains Supplementary material available at https://doi.org/10.1038/s41467-026-70345-y.

