## [Transparent Peer Review File · Nature Communications]

Control of lysosome function by the GTPase-activating protein TBC1D9B and its binding partner TMEM55B

Corresponding Author: Professor Markus Damme

Version 0:

Reviewer comments:

Reviewer #1

(Remarks to the Author)

This manuscript identifies the GTPase-activating protein (GAP) TBC1D9B as a crucial negative regulator of the lysosome-associated small GTPase ARL8B. The authors show that TBC1D9B binds directly to active ARL8B-GTP, stimulates its GTPase activity, and thereby modulates lysosome positioning and the cellular autophagic response. TBC1D9B interacts with the lysosomal membrane protein TMEM55B, which appears to support its localisation and function. Knockout and overexpression studies demonstrate that TBC1D9B and TMEM55B are necessary for proper lysosome dynamics and adaptation to nutrient starvation. This study offers novel mechanistic insight and proposes TBC1D9B as the first known GAP for ARL8B, marking a significant advance in understanding lysosomal regulation. The dataset is of high quality and convincing overall, however the following issues need to be addressed before the manuscript can be recommended for publication.

1. Supplementary figures referenced throughout the text were not available in the submission system and full evaluation of critical claims (e.g., structure predictions, GTPase assays, and proteolysis assays) was impossible.
 2. The mechanism underlying the TMEM55B–TBC1D9B interaction requires clarification. Provide amino acid boundaries for the TBC1D9B fragments used, and reconcile the discrepancy where the N-terminal fragment interacts experimentally but is not predicted to bind by AlphaFold2.
 3. The structural models generated by AF2 should include confidence scores. The key residues mutated in the GAP-inactive TBC1D9B mutant (R559, Y592, Q594) must be clearly annotated and visualised. Triple mutation is likely to affect the structure of the protein (this could also be modelled by AF2). Can authors test single mutants of TBC1D9B to clarify the key residue in GAP function?
 4. The authors should attempt to investigate interactions between endogenous TMEM55B, TBC1D9B, and ARL8B.
 5. It is unclear whether the GAP-deficient TBC1D9B mutant localises to lysosomes to the same extent as the wild-type protein. This should be assessed and quantified.
 6. No statistical analyses are evident in some figures, e.g. in the starvation-induced changes in wild-type cells in Fig. 3b or in cathepsin activity in Fig. 3d. Can authors provide clarification? Significance markers and confidence intervals should be made clearer.
 7. ARL8B GTPase activity in the absence of TBC1D9B should be shown as a baseline control in Fig. 6c.
 8. The observed specificity of TBC1D9B for ARL8B and not ARL8A warrants further exploration, ideally through structural modelling or sequence comparison to elucidate the molecular basis of this selectivity.
 9. Given the high-confidence structural prediction, the authors may consider supporting their model further with molecular dynamics simulations to provide insight into the TBC1D9B–ARL8B interface and its catalytic implications.
 10. Is complex between TBC1D9B, TMEM55B and ARL8B possible? Can this be tested using AF2, MD and experimentally? How does TMEM55B affect GAP function of TBC1D9B?
 11. The authors are encouraged to include autophagy flux assays using reporters such as mRFP-GFP-LC3 or Halo-GFP-LC3, or use bafilomycin A1 treatment to assess LC3-II/p62 turnover.
- Minor:
12. Human gene and protein names should be capitalised consistently throughout the manuscript (e.g., TBC1D9B, ARL8B).
 13. A few figure references are confusing or mislabelled (e.g., "Figure 5a below" in the main text), which should be corrected for clarity.
 14. Domain abbreviations (e.g., GRAM, EF-hand, AID) should be defined at first mention in the text for clarity.
 15. JIP4/TMEM55B introduction/discussion requires additional relevant reference: PMID: 36394115.
 - 16.

Reviewer #2

(Remarks to the Author)

This paper investigates the function of the known lysosomal membrane protein TMEM55B by looking for interaction partners by immunoprecipitation. They find a wide range of proteins enriched in precipitates, including a previously reported interactor JIP4. Amongst the other hits they select TBC1D9B, a member of large family of TBC proteins that act as GAPs on the Rab family of GTPases. They show that varying the levels of TBC1D9B or of TMEM55B alters the distribution of lysosomes. A GTPase Arl8 has been previously shown to be involved in lysosomal positioning and so they test a role for TBC1D9B in regulating Arl8B and find that it can bind the GTP-Arl8B and stimulate its GTPase activity. Finally, they show that the effects of TMEM55B and TBC1D9B on lysosomal positioning require Arl8 to be present.

There is a lot of interest in lysosomal positioning as it controls important aspects of lysosomal function and autophagy. The data in this paper are clearly presented and the text is well written. However, the authors do not make a very compelling case for TBC1D9B either binding TMEM55B or being a major regulator of Arl8. In addition, aspects of the work seem hard to reconcile with previously reported data on TMEM55B or TBC1D9B, some of which is not cited. As such the work does not seem suitable for publication. Below I have explained the major issues with the study, and hope that they will be of use for the authors to revise the paper for resubmission elsewhere.

A) Paralogues. TMEM55B has a paralog TMEM55A that arose during vertebrate evolution and so is likely to have role that is at least partially redundant. Likewise, TBC1D9B has three closely related paralogs, TBC1D9, TBC1D8 and TBC1D8B. Again, these could well be partially redundant as this is often seen with paralogs that arose during vertebrate evolution. However, none these paralogs are mentioned in the paper. It may be that they are expressed at low levels in the cells under investigation, but they should at least have been mentioned, especially as the authors do mention the fact that Arl8 activity is shared by two paralogs Arl8A and Arl8B. Likewise, the authors do not mention that TMEM55B has the gene name of PIP4P1, reflecting reports that it has phosphatase activity on phosphatidylinositol phosphates. They may not believe this activity is correct, but again the gene name and this past work should have been mentioned and a brief reason stated for why they feel it can be ignored.

B) Mouse KOs. The International Mouse Phenotyping Consortium (IMPC) has generated KO mice lacking TMEM55B and TBC1D9B. The mice are viable and fertile and show only a small number of subtle phenotypes, such as reduced startle reflex or hyperactivity. This seems very hard to reconcile with these two proteins being major regulators of Arl8 activity given that mice lacking Arl8 activity (through loss of BORC) show embryonic lethality. This is not mentioned or discussed in the paper.

C) TMEM55B binding TBC1D9B. The authors identify TBC1D9B as an interactor of TMEM55B by doing an IP of over-expressed GFP-TMEM55B. This brought down a wide range of proteins mostly from other organelles such as ER exit sites. Of these >25 proteins TBC1D9B was selected for follow up. The authors show that over-expressed HA-tagged TBC1D9B co-precipitates with over-expressed GFP-TMEM55B. IPs of two over-expressed proteins are notoriously unreliable as both may be a sticky due to lacking normal interaction partners, and so usually one or other is examined at native levels using antibodies to the endogenous protein. The authors attempt to map the region of TBC1D9B that binds to GFP-TMEM55B, and are unable to map it to a specific region which is again indicative of non-specific binding. Finally, AlphaFold does not predict that the two proteins interact, and the interaction was not detected when GFP-TMEM55B expressed at native levels was precipitated by the OpenCell project. Even if the co-ip of TMEM55B and TBC1D9B is physiologically relevant, the authors do not show that it is direct, nor do they discuss the relationship of the interaction with the various others reported for TMEM55B (JIP4, RILPL1, NEDD4 etc) – ie are mutually exclusive.

D) Effect of TBC1D9B on lysosomal positioning. The authors examine the effect on lysosomal localization of over-expressing or removing TMEM55B and/or TBC1D9B. Previous studies have already shown the TMEM55B has an effect, and the authors argue that this is mediated through TBC1D9B, presumably by inactivating Arl8B via its GAP activity. However, this is hard to interpret as Arl8B is known to mediate the recruitment to lysosomes of both kinesin (via PLEKHM2) and dynein (via RUFY3/4). Thus, altering levels of Arl8B activity might be expected to affect the speed rather than the overall distribution of lysosomes.

E) TBC1D9B GAP activity on Arl8B. To determine if Arl8B is involved in the effects of TMEM55B and TBC1D9B, the authors use TurboID on Arl8A to look to see what proteins are close to it. The problem with this approach is that this will label proteins that are not directly binding to Arl8B but are simply on the lysosomal surface with Arl8B. The control that the authors use of a cytosolic TurboID is not suitable as it is not on the lysosomal surface. Having found that the known lysosomal protein TMEM55B is close to the lysosomal GTPase Arl8B, along with TBC1D9B, they then test direct binding and find evidence for TBC1D9B binding to the GTP-bound form of Arl8B. However, this could simply reflect it being an effector of Arl8B rather than its GAP. Moreover, the interaction is only seen with Arl8B rather than the closely related Arl8A, which is surprising, especially as there is good evidence that the two Arl8 paralogs are redundant and so any major regulators would be expected to act on both. They test directly GAP activity of TBC1D9B using GST-Arl8B and detect what they refer to as “low GAP activity” which “may be stimulated in vivo by membrane lipids or post-translational modifications” but this latter is just speculation.

TBC1D9B having Arl8B GAP is also hard to reconcile with Arl8B-GFP being still on lysosomes when TBC1D9B-HA is over-

expressed (Figure 5F). Indeed, there seems to be more Arl8B-GFP on lysosomes in the presence of TBC1D9B overexpression than without (compare Figure 5F with Figure 6D).

Overall, the data in the paper do not support many of the conclusions that the authors wish to make. It may be that Arl8B recruits TBC1D9B to lysosomes as an effector – perhaps to inactivate its known substrate Rab11, but the data do not support the conclusion that TMEM55B binds directly to TBC1D9B, nor the conclusion that TBC1D9B has GAP activity on Arl8B.

Reviewer #3

(Remarks to the Author)

The manuscript 'Control of lysosome function by the GTPase-activating protein TBC1D9B and its binding partner TMEM55B' by Duhay and Tian et al, describes a number of interesting findings with regards to the role of the protein TBC1D9B in lysosomal function. The study identifies TBC1D9B as a key negative regulator of the lysosomal small GTPase Arl8b, showing that it binds Arl8b-GTP and promotes its inactivation. In addition, the study shows that loss of TBC1D9B, or its partner TMEM55B, disrupts lysosome positioning. The authors conclude that the loss of TBC1D9B affects autophagic flux, an effect that is dependent on Arl8b.

The finding that TBC1D9B has GAP activity towards Arl8b, is novel, and will be of interest to a broad community of researchers interested in organelle biology, membrane trafficking, and cellular nutrient sensing. Generally, the manuscript is well-written, the methodology is appropriate and most of the conclusions drawn are supported by the data. However, some conclusions are not supported by sufficient evidence and would require additional experiments to be conducted. It is recommended that the authors revise the current manuscript to address a few main concerns, as well as a list of minor items, appended below.

Main Items

1) Figure 3c-d shows an impaired increase in Cathepsin D activity in response to starvation in TBC1D9B and TMEM55B KO lines. However, it is not clear why this is, or even how the increased activity in WT cells is working. The paper cited (which is from one of the corresponding authors of this manuscript), Ebner M. et al. 2023, showed a lipid switch at lysosomes that alters lysosome functions including proteolytic activity, and blocking this lipid switch alters lysosome position. Another study also cited here, Johnson D.E. et al. 2016, shows that peripheral lysosomes are more alkaline – this could also stand to reduce Cathepsin D activity. Or, do the authors think that 6h starvation is causing significant transcriptional changes that boost Cathepsin D activity?

The rationale and result should be discussed in more detail.

2) In the text relating to Figure 4, I would hesitate to conclude anything about lysosome “dynamics”. All the data is based on images of fixed cells. Dynamics implies that one would be measuring lysosomal speed or velocity, motility tracking to determine if they are undergoing motor-based transport, or lysosomal morphology changes/membrane remodeling such as tubulation, fusion, fission.

Live-cell imaging experiments would need to be performed to compute some of the above parameters and make conclusions about lysosome dynamics. Indeed, such additional data would strengthen the manuscript. However, at the very least, Figure 4 (and the corresponding section in the text) should be labeled and discussed as the effects on lysosome clustering/positioning upon overexpression of TBC1D9B and TMEM55B.

3) What is the knockdown efficiency of siTBC1D9B in Figure 7c-d? Western blot or, if there is no suitable antibody, qPCR, should be conducted and shown.

4) The single immunostaining experiment for LC3 is not sufficient to suggest that the lower levels of LC3 observed in the siTBC1D9B condition is due to increased autophagic flux as flux wasn't measured. An autophagy flux assay (for example, Halo-Tag Flux assay PMID: 36095089) should be performed, if the authors wish to make these conclusions.

On that note, how does this result (if the authors do suggest there is increased autophagic flux in siTBC1D9B cells) compare to the findings from Liao et al. 2018 that concluded TBC1D9B promoted autophagic flux and it's silencing reduced autophagic degradation?

Minor Items

-Regarding the callout of Figure 1f in the text: “Conversely, TBC1D9B-GFP was co-immunoprecipitated with TMEM55B (Figure 1f).” This second co-IP is in Figure 1e (right).

-Figure 1f – I would recommend moving all the red labels to one side (i.e. right) of the drawn constructs as it is a bit difficult, at first, to determine which label goes with which fragment

-Supplemental Figure 1c: What is the asterisk in the HA blot referring to? Legend says the asterisk shows cleaved GFP, but it is also shown in the HA blot.

-Since OrgaMapper is used throughout to measure lysosome positioning, the authors should give a brief description (a couple of sentences) about how it works, in addition to the reference to the bioRxiv preprint that is already there. For example, readers will wonder if cell size is taken into consideration in this analysis since the TMEM55B and TBC1D9B KOs are obviously larger than WT. On this note, this phenotype should be mentioned (if it is real), and followed up by a statement on how OrgaMapper normalizes organelle positioning to cell size (i.e. diameter?) and is therefore a suitable method. It would help the readers as they could have doubts about this and then having to go into the OrgaMapper preprint (as I did) to see whether this method controls for cell size, is a minor inconvenience.

-The statement: "Rab11-positive recycling endosomes were also partially redistributed to the cell periphery in TBC1D9B KO cells, consistent with an earlier study 34" should be accompanied by a callout to Supplemental Figure 2 (Rab11 staining)

-Figure 2d should have a small legend for the schematic, as shown in Figure 4b.

-Callout of Figure 2e in the text is missing a parenthesis

-Figure 2e – The authors should show the individual channels in the whole-cell image. It looks, to this reviewer, like TBC1D9B-HA may have an ER localization, as well as some puncta that overlap with LAMP2. This ER localization is somewhat corroborated in Figure 2f, yet there is no discussion of this additional subcellular localization in cells. Is it also in the ER? If so, do the authors think this is real/expected? Or possibly a virtue of overexpression? An IF in which the ER is co-stained, would determine if any TBC1D9B signal is there.

It is good that the overexpression is done in the background of the KO cells rather than WT to reduce overexpression-induced artefacts, however, TBC1D9B may be in the ER and this is not discussed. It is concluded that it has a cytosolic localization (and on some lysosomes) but the cytosolic localization is not convincing.

-It is not obvious by eye in Figure 2f that the GAP activity-deficient TBC1D9B mutant is unable to rescue the lysosome dispersal phenotype. On the contrary, it looks like a partial rescue. Is this a matter of choosing a more representative image?

-Figure 2g quantification should have the WT condition quantified as well because the experiment is measuring a "rescue" of lysosome positioning. This rescue can be best assessed by seeing the quantification of WT cells.

-In this statement on pg 6: "As expected, starvation-induced marked relocalization of lysosomes towards the microtubule organizing center in wildtype cells." there should be no hyphen, just "starvation induced..."

-Is the TBC1D9B KO (fed) image in Figure 3a representative? The lysosomes do not look as peripheral as in Figure 2b and Figure 2f.

-Regarding Figure 3 c-d: The SiR lysosome signal in the fed condition, at least from the images in Figure 3c, doesn't seem to be the same between the genotypes (i.e. TMEM55B KO seems to have very diminished signal). Is this due to the extreme dispersal of lysosomes just making it difficult to see this signal by eye? SiR lysosome signal should be quantified in this condition alone across the different genotypes (fed) and perhaps placed into Supplemental.

-Supplemental Figure 3: Was the DQ-BSA allowed to co-endocytose with a pH- and protease-insensitive fluorescent probe, such as Alexa555-dextran to normalize for amount of endocytic uptake? It seems from the figure legend that cells were given DQ-BSA following the treatments (fed/starved). Thus, the difference in fluorescence intensity of DQ-BSA in the KOs could also be due to altered endocytosis. Authors are advised to either perform this experiment with an endocytosis control (as suggested above), or at the very least, mention this limitation of their current approach in the text.

-Figure 4c should be accompanied by the Pearson's correlation coefficient in TBC1D9B KO + TBC1D9B-HA staining with LAMP2. Since this result is not discussed, it is unclear if the Pearson's value shown in Figure 4c actually means loss of TBC1D9B from lysosomes and into the cytosol in the o/e condition. The result therefore seems arbitrary.

-It is not clear why Figure 5 a/b was done in iNeurons? It is also not clear why an "unbiased" approach needed to be taken here. The authors had just identified TBC1D9B as a regulator of lysosome positioning and its GAP activity was important for this. The next logical step would be to look, in a targeted manner, at the lysosomal small GTPases known to couple to motors - Arl8 and Rab7. An explanation about the experimental rationale here would be useful.

-In the statement "A GAP-activity-defective TBC1D9B mutant (TBC1D9BRYQ/AAA) (see Fig. 5a below) retained the ability to interact with Arl8b-GFP (Q75L) (Figure 5d)." What is the first callout "(see Fig 5a below)" referring to?

-Why does the overexpression of WT Arl8b (Figure 5f) show a lower Pearson's correlation coefficient for Lamp2/TBC1D9B? From the images it looks like there's good LAMP2/TBC1D9B overlap in the WT Arl8b condition.

-Figure 6d – The images for TMEM55B KO do not seem to show any more increase in Arl8b recruitment to LAMP2 compartments than WT. Are these images representative?

-In this statement on pg 9: "Given the defects of TBC1D9B-depleted cells with respect to the lysosomal starvation response (compare Figure 2), we monitored the presence of LC3 positive autophagosomes in cells depleted of TBC1D9B, Arl8a/b, or both proteins." In the callout, do the authors mean compare Figure 3?

-I am assuming from the discussion of Figure 7, that Figure 7c and d are showing cells under starvation condition? If so, this is not labeled in the micrographs (Fig 7c), the quantification (Fig 7d) or the figure legend. Is the quantification in Figure 7d referring to fold change in starved condition over fed condition? Is the WT siTBC1D9B quantification significantly different from WT scr? Also, check the labels in Figure 7d, they say TBC1B9D. And, as stated in main item #3, KD efficiency needs to be shown here.

-With regards to this statement on pg 9: "Third, we show that TBC1D9B directly and selectively associates with active Arl8b-GTP in living cells and in vitro (Figure 4a-e)." Are the authors referring to Figure 5?

-With regards to this statement on pg 10: "In preliminary experiments in HeLa cells overexpressing TMEM55B, we failed to detect any overt effects on TBC1D9B localization (Supplementary Figure 6a)." Supplemental Figure 6a doesn't show TBC1D9B staining, only LAMP2 and the TMEM55B-GFP expression in WT and TBC1D9B KO cells.

-In methods, change "Mutagenesis of TBCD9B" to "Mutagenesis of TBC1D9B"

Version 1:

Reviewer comments:

Reviewer #1

(Remarks to the Author)

The authors have addressed my comments sufficiently and the manuscript can be recommended for publication.

Reviewer #2

(Remarks to the Author)

This is a revised version of a paper that investigates the lysosomal membrane protein TMEM55B and finds an interaction with TBC1D9B which the authors then argue acts as a GAP for the lysosomal GTPase Arl8B, in addition to its previously reported role as a GAP for Rab11.

My review of the original version of the manuscript I noted the potential interest in the regulation of lysosomal positioning but expressed concerns about the strength of the evidence for TMEM55B binding directly to TBC1D9B and for the latter having GAP activity on Arl8B. In their rebuttal the authors state on several occasions that they "respectfully disagree" with my comments, but nonetheless they have added a significant amount of new data to strengthen these central conclusions of the paper. Overall, this has been done to a high standard and the paper seems significantly improved. As such I am happy to "respectfully" recommend acceptance. There are a couple of minor issues that would need addressing before publication:

a) The authors cite a recent study describing the binding between TMEM55B and a different partner, a study which synergises well with their own work. However, this paper is cited as Waschbusch et al 2025, but it is in fact from 2026.

b) As the authors discuss, there are likely to be multiple regulators of Arl8B and its paralog Arl8A, and in this context they should cite a recent preprint that reports that DENND6A also has GAP activity toward ARL8B (PubMed ID 41542567).

Reviewer #3

(Remarks to the Author)

Overall, I believe the additional experiments and the corrections the authors have implemented have greatly improved the manuscript and I am satisfied with the authors' response. However, please correct all of the following:

1) The manuscript file contains a duplication of the manuscript. Pages 1-36 is one manuscript without figures and pages 37 (labeled as page 1)-79 is another copy/version of the manuscript with figures. I assume this was a copy/paste error, but I am not sure if the two manuscripts are different in any way?

2) Figure 2e and Supp Fig 2a – It is nice to see that lysosome speed has been computed. However, units of the speed are either incomplete (Figure 2f currently says "Speed>0.1 μm ", do the authors mean speed > 0.1 $\mu\text{m}/\text{sec}$?) or missing (Supp Figure 2 has no units of speed on y-axis or in the legend). Speed should be presented as distance/time.

3) Supp Fig 3d –The significance stars appear shifted to the left of the graph. The authors should check that this is moved to the right place. The legend for Supp Fig 3d doesn't correspond to TBC1D9B KD experiment.

4) Supp Fig 3e – "TBC1D9B fed plus +BafA1", the authors may want to remove the "plus"? Again, the legend needs to be corrected for this panel.

5) Supp Fig 3b – Legend needs correction – refers also to a dextran experiment.

6) Overall for Supp Fig 3: Fix the entire legend as it also mentions a panel (f) and there is no panel f in the figure.

Reviewer #1 (Remarks to the Author):

This manuscript identifies the GTPase-activating protein (GAP) TBC1D9B as a crucial negative regulator of the lysosome-associated small GTPase ARL8B. The authors show that TBC1D9B binds directly to active ARL8B-GTP, stimulates its GTPase activity, and thereby modulates lysosome positioning and the cellular autophagic response. TBC1D9B interacts with the lysosomal membrane protein TMEM55B, which appears to support its localisation and function. Knockout and overexpression studies demonstrate that TBC1D9B and TMEM55B are necessary for proper lysosome dynamics and adaptation to nutrient starvation. This study offers novel mechanistic insight and proposes TBC1D9B as the first known GAP for ARL8B, marking a significant advance in understanding lysosomal regulation. The dataset is of high quality and convincing overall, however the following issues need to be addressed before the manuscript can be recommended for publication.

Response: We thank the reviewer for the lucid summary and these highly positive comments and her/his support to publish our study in Nat. Commun.

1. Supplementary figures referenced throughout the text were not available in the submission system and full evaluation of critical claims (e.g., structure predictions, GTPase assays, and proteolysis assays) was impossible.

Response: We sincerely apologize for the apparent error. In the revised submission, all files are available.

2. The mechanism underlying the TMEM55B–TBC1D9B interaction requires clarification. Provide amino acid boundaries for the TBC1D9B fragments used, and reconcile the discrepancy where the N-terminal fragment interacts experimentally but is not predicted to bind by AlphaFold2.

Response: We agree and have included the amino acid boundaries of the TBC1D9B fragments in the figure and results section for clarity. We want to highlight that during the revision of this manuscript, another lab fully independently found the interaction between TMEM55B and TBC1D9B in an unbiased interactomics screen for TMEM55B (in the presence of active Rab8a and LRRK2) and confirmed the interaction by immunoprecipitation. Waschbüsch et al. identified an interaction motif/surface in TMEM55B-interactors (TMEM55B-binding motif (TBM)) that is also present in TBC1D9B (around amino acid position 940); these data are in good agreement with our mapping of the interaction surface (with constructs including the TBM motif, e.g., C-terminus, TBC-domain+EF-hand). The fact that the N-terminus of TBC1D9B can also associate with TMEM55B might reflect an additional interaction site independent of the TBM motif. Finally, it is important to consider that while AlphaFold may serve as a valuable predictor of protein-protein interactions, experimental data are required to verify such predictions, and the actual interaction site(s) may differ from the computational model.

3. The structural models generated by AF2 should include confidence scores. The key residues mutated in the GAP-inactive TBC1D9B mutant (R559, Y592, Q594) must be clearly annotated and

visualised. Triple mutation is likely to affect the structure of the protein (this could also be modelled by AF2). Can authors test single mutants of TBC1D9B to clarify the key residue in GAP function?

Response: We added confidence scores (pLDDT) and the PAE matrix to the new Supplemental Figure 5a. Moreover, key residues in TBC1D9B (R559, Y592, Q594) in the AF3 model of the TBC1D9B-ARL8B complex are now highlighted as red spheres in the main Figures 6a and b. Given the amount of experimental effort invested into the revisions of this manuscript and the laborious procedure for producing recombinant TBC1D9B, we have not been able to assess the effects of single mutants of TBC1D9B.

4. The authors should attempt to investigate interactions between endogenous TMEM55B, TBC1D9B, and ARL8B.

Response: Our initial attempts to analyze the interaction between endogenous TBC1D9B and TMEM55B suffered from the weak affinity of the available TBC1D9B antibodies. To nonetheless address the referee's suggestion, we have used CRISPR/Cas9 to generate KI cell lines endogenously expressing HA-tagged TBC1D9B (new Figure 1f). We now demonstrate in the new Figure 1g complex formation between endogenous TBC1D9B-HA and TMEM55B. These data are further supported by demonstrating a direct interaction between recombinant GST-TMEM55B and TBC1D9B-GFP as shown in the new Figure 1h. At our level of sensitivity, we were unable to detect ARL8B in TMEM55B immunoprecipitates, likely because of the transient nature of the TMEM55B-TBC1D9B-ARL8 complex. Finally, in the new Figure 5e, we demonstrate the formation of a tripartite TMEM55B-TBC1D9B-ARL8 complex in vitro using purified recombinant proteins.

5. It is unclear whether the GAP-deficient TBC1D9B mutant localises to lysosomes to the same extent as the wild-type protein. This should be assessed and quantified.

Response: We agree and included a quantification/ Pearson correlation between wild-type and mutant TBC1D9B^{RYQ/AAA} and lysosomes/LAMP2 by confocal microscopy imaging (new Figure 4d).

6. No statistical analyses are evident in some figures, e.g. in the starvation-induced changes in wild-type cells in Fig. 3b or in cathepsin activity in Fig. 3d. Can authors provide clarification? Significance markers and confidence intervals should be made clearer.

Response: We have carefully revised all legends and manuscript text to clearly indicate statistical significance, including the respective figures noted by the referee. We assume the referee meant figure 7 (ARL8B knockdown effect on positioning and LC3 +/-ARL8B) and added statistics for the respective panels.

7. ARL8B GTPase activity in the absence of TBC1D9B should be shown as a baseline control in Fig. 6c.

Response: In response to this referee and referee #2 we have gone through great biochemical efforts to obtain purified WT and GAP-defective mutant TBC1D9B from insect cells (rather than immunoprecipitated protein) and have assayed its ability to promote ARL8 GTPase activity in vitro. Our new data shown in Fig. 6c and Suppl. Fig. 5b-e clearly demonstrate that TBC1D9B acts as a GAP for ARL8B with an efficacy comparable to that for its known substrate Rab11. The baseline of ARL8B GTPase activity is now clearly visible in Figure 6c.

8. The observed specificity of TBC1D9B for ARL8B and not ARL8A warrants further exploration, ideally through structural modelling or sequence comparison to elucidate the molecular basis of this selectivity.

Response: We were also puzzled to see this. In the revised manuscript, we now provide a sequence comparison between ARL8A and ARL8B (see new Suppl. Fig. 4e) and discuss potential explanations

based on the subtle differences in primary sequence between both proteins in the revised discussion section of our paper.

9. Given the high-confidence structural prediction, the authors may consider supporting their model further with molecular dynamics simulations to provide insight into the TBC1D9B– ARL8B interface and its catalytic implications.

Response: Unfortunately, we do not have the technical know-how or capacity to conduct MD simulations in our groups. We agree that further structural studies will be an interesting avenue for future research.

10. Is complex between TBC1D9B, TMEM55B and ARL8B possible? Can this be tested using AF2, MD and experimentally? How does TMEM55B affect GAP function of TBC1D9B?

*Response: We thank the referee for this interesting point and suggestion. In response, we have conducted additional experiments and demonstrate in the **new Figure 5e** the formation of a tripartite TMEM55B-TBC1D9B-ARL8 complex using purified recombinant proteins.*

Due to a lack of a sufficient amount of pure soluble TMEM55B we have not been able to study the effect of TMEM55B on the GAP activity of TBC1D9B in vitro. We hope that the editor and referee#1 agree that such studies go beyond the scope of the present manuscript.

11. The authors are encouraged to include autophagy flux assays using reporters such as mRFP-GFP-LC3 or Halo-GFP-LC3, or use bafilomycin A1 treatment to assess LC3-II/p62 turnover.

*Response: We thank the reviewer for this suggestion that we have addressed in two ways: First, we show that depletion of TBC1D9B results in a moderate accumulation of the ubiquitin adaptor and autophagic substrate p62 (**new Suppl. Fig. 3e**). In addition, we have used Halo-LC3 expressing cell lines kindly provided by Prof. Mizushima to monitor autophagic flux directly. These data shown in the **new Figure 3e**, further confirm the reduction of autophagic flux assayed by auto-lysosomal cleavage of Halo-LC3.*

Minor:

12. Human gene and protein names should be capitalised consistently throughout the manuscript (e.g., TBC1D9B, ARL8B).

Response: We changed the gene-/protein names accordingly to capitalised letters.

13. A few figure references are confusing or mislabelled (e.g., "Figure 5a below" in the main text), which should be corrected for clarity.

Response: We corrected the mislabelling.

14. Domain abbreviations (e.g., GRAM, EF-hand, AID) should be defined at first mention in the text for clarity.

Response: We have written out the abbreviations in full when they are first mentioned. EF-hand is not an abbreviation but stands for "helix E and helix F" in EF-hand containing proteins. For clarity, we left this explanation out.

15. JIP4/TMEM55B introduction/discussion requires additional relevant reference: PMID: 36394115.

Response: We added the reference according to the suggestion.

Reviewer #2 (Remarks to the Author):

This paper investigates the function of the known lysosomal membrane protein TMEM55B by looking for interaction partners by immunoprecipitation. They find a wide range of proteins enriched in precipitates, including a previously reported interactor JIP4. Amongst the other hits they select TBC1D9B, a member of large family of TBC proteins that act as GAPs on the Rab family of GTPases. They show that varying the levels of TBC1D9B or of TMEM55B alters the distribution of lysosomes. A GTPase ARL8 has been previously shown to be involved in lysosomal positioning and so they test a role for TBC1D9B in regulating ARL8B and find that it can bind the GTP-ARL8B and stimulate its GTPase activity. Finally, they show that the effects of TMEM55B and TBC1D9B on lysosomal positioning require ARL8 to be present.

There is a lot of interest in lysosomal positioning as it controls important aspects of lysosomal function and autophagy. The data in this paper are clearly presented and the text is well written.

Response: We thank the reviewer for highlighting the importance and high general interest of the topic as well as for stressing the clarity of the manuscript text.

However, the authors do not make a very compelling case for TBC1D9B either binding TMEM55B or being a major regulator of ARL8. In addition, aspects of the work seem hard to reconcile with previously reported data on TMEM55B or TBC1D9B, some of which is not cited. As such the work does not seem suitable for publication. Below I have explained the major issues with the study, and hope that they will be of use for the authors to revise the paper for resubmission elsewhere.

Response: We respectfully disagree with the referee's general assessment but take his/ her concerns very seriously and have addressed them by conducting a series of additional experiments and by textual changes and explanations.

A) Paralogues. TMEM55B has a paralog TMEM55A that arose during vertebrate evolution and so is likely to have role that is at least partially redundant. Likewise, TBC1D9B has three closely related paralogs, TBC1D9, TBC1D8 and TBC1D8B. Again, these could well be partially redundant as this is often seen with paralogs that arose during vertebrate evolution. However, none these paralogs are mentioned in the paper.

It may be that they are expressed at low levels in the cells under investigation, but they should at least have been mentioned, especially as the authors do mention the fact that ARL8 activity is shared by two paralogs ARL8A and ARL8B.

Response: We now mention the TMEM55A paralog in the introduction and the TBC1D9B paralogs in the discussion of the paper, although we also note that their sequences are in fact substantially different from that of TBC1D9B. Whether or not these proteins serve redundant functions, in our view, cannot be judged from mere sequence comparison or speculation. To address this point at least partially experimentally, we generated TBC1D9/TBC1D9B double knockout cells. TBC1D9 knockout cells show a mild dispersion of lysosomes to the periphery and this phenotype is aggravated in double knockout cells lacking both TBC1D9 and TBC1D9B, suggesting that TBC1D9 and TBC1D9B might fulfil a (partially) redundant function (new Suppl. Fig. 2c). We hope the reviewer acknowledges the complexity given the various different paralogs of the proteins involved (i.e. ARL8A/ARL8B, TMEM55A/TMEM55B, TBC1D9/TBC1D9/TBC1D8/TBC1D8B) that has precluded analysis of all possible combinations of protein isoforms experimentally.

We acknowledge this in the discussion by stating:

"...Several other open questions emanate from our study. For example, it is surprising that TBC1D9B appears to selectively act as a negative regulator of ARL8B function but not of the closely related ARL8A isoform, in spite of the profound sequence similarity(identity of both isoproteins (~91% identity) 18. A prediction from our work, therefore, is that other GAP proteins must exist that act to physiologically control ARL8 activity in cells and tissues. Such GAP proteins could conceivably act selectively on ARL8a or on both ARL8 isoforms or exhibit cell and tissue specificity. Candidates are

other family members of the TBC family, including the TBC1D9B paralogs TBC1D9, TBC1D8, and TBC1D8B. Our experiments in TBC1D9/TBC1D9B double KO cells suggest that they share similar functions. Adding to the overall complexity, TMEM55B has a close paralog: TMEM55A. TMEM55A does not bind JIP4 31, suggesting some degree of specificity, and its contribution to the TBC1D9B/ARL8B axis remains to be determined.”

Likewise, the authors do not mention that TMEM55B has the gene name of PIP4P1, reflecting reports that it has phosphatase activity on phosphatidylinositol phosphates. They may not believe this activity is correct, but again the gene name and this past work should have been mentioned and a brief reason stated for why they feel it can be ignored.

Response: This is another fair point that we were happy to address. We note that current experimental data (including the recently solved crystal structure) do NOT support a lipid phosphatase for TMEM55B and cited the corresponding references (Ungewickell et al PNAS 2005; Willet et al Nat Commun 2017; Waschbüsch et al Structure 2025).

Introduction

“TMEM55B is a 284-residue lysosomal protein and has been annotated as a phosphatidylinositol phosphatase (also named as PIP4P1) 32; However, TMEM55B lacks sequence similarities to other lipid phosphatases, and in vitro activity as a lipid phosphatase could not be validated in a later study 33. The crystal structure of TMEM55B also does not support its function as a phosphatase, but has shown that TMEM55B represents a platform for the binding of various interaction partners 34.”

B) Mouse KOs. The International Mouse Phenotyping Consortium (IMPC) has generated KO mice lacking TMEM55B and TBC1D9B. The mice are viable and fertile and show only a small number of subtle phenotypes, such as reduced startle reflex or hyperactivity. This seems very hard to reconcile with these two proteins being major regulators of ARL8 activity given that mice lacking ARL8 activity (through loss of BORC) show embryonic lethality. This is not mentioned or discussed in the paper.

Response: We respectfully disagree with the interpretation of the referee. First, TBC1D9B, according to our data (Suppl. Fig. 4), selectively acts on ARL8B but not on ARL8A. Hence, phenotypes resulting from the loss of TBC1D9B are expected to be milder than those resulting from the complete loss of ARL8A and ARL8B. We further note that no complete loss of function models for ARL8A and ARL8B (i.e. DKO mice) have been reported in the literature. Second, the quoted loss of BORC in our view cannot be regarded as equal to double loss of ARL8A and ARL8B. In our earlier study in Science (Rizalar et al 2023), we demonstrated that loss of BORC in either human neurons or D. melanogaster exhibits phenotypes that are much milder than those elicited by double loss of ARL8A and ARL8B. In the same study, we have also shown that the phenotype of a single ARL8B knockout is comparably mild with respect to axonal transport of either lysosomes or synaptic vesicle precursors. Third, loss of TBC1D9B is predicted to result in a gain of ARL8B function. How this translates phenotypically at the level of an animal (i.e. mice) to our knowledge has not been addressed. Hence, no comparison to the phenotype of ARL8 or BORC KO mice can be made.

In response, we have edited the discussion section of our manuscript to make these points clear.

In conclusion, we do not see our data or interpretation to clash with any previous findings in the literature nor to be inconsistent with the points voiced by referee #2. In all fairness, we hope that referee #2 agrees to this!

C) TMEM55B binding TBC1D9B. The authors identify TBC1D9B as an interactor of TMEM55B by doing an IP of over-expressed GFP- TMEM55B. This brought down a wide range of proteins mostly from other organelles such as ER exit sites. Of these >25 proteins TBC1D9B was selected for follow up. The authors show that over-expressed HA-tagged TBC1D9B co-precipitates with over- expressed GFP- TMEM55B. IPs of two over-expressed proteins are notoriously unreliable as both may be a sticky due to lacking normal interaction partners, and so usually one or other is examined at native levels using

antibodies to the endogenous protein. The authors attempt to map the region of TBC1D9B that binds to GFP-TMEM55B, and are unable to map it to a specific region which is again indicative of non-specific binding. Finally, AlphaFold does not predict that the two proteins interact, and the interaction was not detected when GFP-TMEM55B expressed at native levels was precipitated by the OpenCell project.

Response: We respectfully disagree. We added experimental evidence strongly supporting the direct interaction between TMEM55B and TBC1D9B:

We have added an affinity chromatography experiment using recombinant GST-TMEM55B and TBC1D9B-GFP, validating the interaction in vitro and showing that its a direct interaction (new Fig. 1h). We agree with the reviewer that overexpression (abd in particular double-overexpression) can often lead to non-specific interactions; therefore, we generated TBC1D9B-HA knockin cells and validated the interaction fully at the endogenous level (new Fig. 1g). Together, these data strongly support the specific interaction between TMEM55B and TBC1D9B.

We want to highlight that during the revision of this manuscript, another lab fully independently found the interaction between TMEM55B and TBC1D9B in an unbiased interactomics screen for TMEM55B (in the presence of active Rab8a and LRRK2) and confirmed the interaction by immunoprecipitation. Notably, there is a surprisingly high overlap between the identified candidates in their screen, strongly supporting the validity of our interactomics data; interactors like RNF213 and BIRC6 were among their top hits. Importantly, Waschbüsch et al. identified an interaction motif/surface in TMEM55B-interactors (TMEM55B-binding motif (TBM)) that is also present in TBC1D9B (around amino acid position 940) (Waschbüsch et al Structure 2025); these data are in good agreement with our mapping of the interaction surface (with constructs including the TBM motif (C-terminus, TBC-domain+EF-hand) showing interaction. The fact that the N-terminus also binds might reflect an additional interaction site independent of the TBM motif. We refer to Waschbüsch et al. for a detailed investigation of the structural basis of the TMEM55B-TBC1D9B interaction, including the AlphaFold model.

Finally, we do not believe that the fact that the interaction was not found in a specific high-throughput screen (OpenCell) is a strong argument. Such screens never reflect the complete interactome. We would also like to point out once again that the interaction between TMEM55B and TBC1D9B was found multiple times in BioPlex, a very reliable interactome dataset (Huttlin et al Nature 2017; Huttlin et al Cell 2021).

Overall, we think there are multiple strong experimental findings from our work and others supporting the idea that the (transient) interaction between TMEM55B and TBC1D9B is real and physiologically relevant.

Even if the co-ip of TMEM55B and TBC1D9B is physiologically relevant, the authors do not show that it is direct, nor do they discuss the relationship of the interaction with the various others reported for TMEM55B (JIP4, RILPL1, NEDD4 etc) – ie are mutually exclusive.

Response: We agree with the referee that, from the experimental data presented in the previous manuscript, an indirect interaction cannot be excluded. As already explained above, we included a pull-down experiment with rec. GST-TMEM55B and rec. TBC1D9B-GFP, validating the DIRECT interaction in an in vitro environment(new Fig. 1h) . We extended the discussion regarding the interaction of TMEM55B with various other interactors/clients.

“...Conversely, we do not yet understand how complex formation between TMEM55B and its various interactors, particularly its association with the dynein/ kinesin adaptor JIP4^{30,31} are regulated, e.g., by phosphorylation⁴⁷. For example, whether TMEM55B can bind to JIP4 and TBC1D9B simultaneously or whether both proteins compete for TMEM55B remained unclear from our study. During the revision of this manuscript, a crystal structure of TMEM55B was published, and TMEM55B was identified as a central hub for adaptor recruitment on lysosomes, binding various clients via a TMEM55B-binding motif (TBM)⁴⁸. These data strongly suggest that different interactors compete for

binding by a hitherto unknown regulatory mechanism. Preliminary experiments suggest that TBC1D9B is dispensable for the reported perinuclear clustering phenotype induced by TMEM55B overexpression (Supplementary Figure 6b-d), which, instead, may reflect its association with JIP4 as reported earlier, supporting a competitive binding of different clients³¹.”.

D) Effect of TBC1D9B on lysosomal positioning. The authors examine the effect on lysosomal localization of over-expressing or removing TMEM55B and/or TBC1D9B. Previous studies have already shown the TMEM55B has an effect, and the authors argue that this is mediated through TBC1D9B, presumably by inactivating ARL8B via its GAP activity. However, this is hard to interpret as ARL8B is known to mediate the recruitment to lysosomes of both kinesin (via PLEKHM2) and dynein (via RUFY3/4). Thus, altering levels of ARL8B activity might be expected to affect the speed rather than the overall distribution of lysosomes.

*Response: A large body of literature including series of papers from the Juan Bonifacino (Pu et al Dev Cell 2015; Guardia et al Cell Rep 2016 etc), Sean Munro (Rosa-Ferreira & Munro Dev Cell 2011), Kang Shen (Klassen et al Neuron 2010; Wu et al Neuron 2013), David Rubinsztein (Korolchuk et al Nat Cell Biol 2011), our own (Vukoja et al Neuron 2018; Rizalar et al Science 2013), and many other laboratories have demonstrated in numerous cell types and across species that the NET RESULT of ARL8 inactivation is the perinuclear accumulation of lysosomes and related organelles, whereas ARL8 hyperactivity has been demonstrated to cause lysosome dispersion. In contrast, studies on the recruitment of dynein via RUFY3/4 (Teren-Kaplan et al Nat Commun 2022; Kumar et al Nat Commun 2022) do not contain evidence that ARL8B is a major recruitment factor for dynein on lysosomes (instead, RUFYs may act on endosomes). That said, we have measured the kinetics of lysosome movement in WT and TBC1D9B loss-of-function models experimentally by additional live cell imaging experiments. In the **new Figure S2a**, we demonstrate that the average speed of lysosome movement is increased in TBC1D9B KO cells, a phenotype that is rescued by re-expression of TBC1D9B-eGFP. The fraction of moving lysosomes is increased in both TMEM55B and TBC1D9B KO cells (**new Figure 2e**).*

E) TBC1D9B GAP activity on ARL8B. To determine if ARL8B is involved in the effects of TMEM55B and TBC1D9B, the authors use TurboID on ARL8A to look to see what proteins are close to it. The problem with this approach is that this will label proteins that are not directly binding to ARL8B but are simply on the lysosomal surface with ARL8B. The control that the authors use of a cytosolic TurboID is not suitable as it is not on the lysosomal surface. Having found that the known lysosomal protein TMEM55B is close to the lysosomal GTPase ARL8B, along with TBC1D9B, they then test direct binding and find evidence for TBC1D9B binding to the GTP-bound form of TBC1D9B. However, this could simply reflect it being an effector of ARL8B rather than being its GAP. Moreover, the interaction is only seen with ARL8B rather than the closely related ARL8A, which is surprising, especially as there is good evidence that the two ARL8 paralogs are redundant and so any major regulators would be expected to act on both. They test directly GAP activity of TBC1D9B using GST-ARL8B and detect what they refer to as “low GAP activity” which “may be stimulated in vivo by membrane lipids or post-translational modifications” but this latter is just speculation.

*Response: Again, we respectfully disagree with the referee’s interpretation and criticism of our data. It is well established that conceptually all GAPs for small GTPases associate preferentially with the GTP-bound version of the protein and, thus, behave indistinguishably from other types of effector proteins. We thus do not see a contradiction with any of our data. Second, we agree that the selective association of TBC1D9B with ARL8B but not ARL8A is surprising. However, such regulatory diversity is certainly conceivable. Our experimental data in Suppl. Fig. 4 are quite clear-cut. In the revised manuscript, we have also provided a sequence comparison between ARL8A and ARL8B (in the **new suppl. fig. 4c**) and discuss potential differences between the two proteins in the discussion section.*

Second, and more importantly, we would like to reiterate that the TBC1D9B GAP activity we observe is specific as a point mutant, in which the critical R finger has been mutated, is inactive in vitro and in

cell-based functional assays in vivo. To provide additional support for the claim that TBC1D9B negatively controls ARL8B function, we have conducted a further series of experiments:

*- We have gone through great biochemical efforts to obtain purified WT and GAP-defective mutant TBC1D9B from insect cells (rather than immunoprecipitated protein) and have assayed its ability to promote ARL8 GTPase activity in vitro. Our **new data shown in Fig. 6c and Suppl. Fig. 5b-e** clearly demonstrate that TBC1D9B acts as a GAP for ARL8B with an efficacy comparable to that for its known substrate RAB11. We hope that the referee will agree that these data compellingly support our model that TBC1D9B is a negative regulator of ARLB function by promoting GTP hydrolysis. We do not question its previously suggested role in regulating RAB11.*

*- As an orthogonal approach, we have depleted cells of RAB11A/B or ARL8A/B to determine whether the effects of TBC1D9B loss on lysosome position are preserved under these conditions. We demonstrate in **the new Suppl. Fig. 4a,b** shows that TBC1D9B loss causes lysosome dispersion in WT and in RAB11A/b knockdown cells but not in cells depleted of ARL8B (as shown in main Fig. 7a,b). These data rule out the possibility that TBC1D9B exhibits its cellular effects indirectly via altering the nucleotide status of Rab11.*

Finally, we have modified the discussion section of our paper to allow for the possibility that TBC1D9B acts on additional GTPases (as one can NEVER rule out such a possibility, no matter what we do).

TBC1D9B having ARL8B GAP is also hard to reconcile with ARL8B-GFP being still on lysosomes when TBC1D9B-HA is over-expressed (Figure 5F). Indeed, there seems to be more ARL8B-GFP on lysosomes in the presence of TBC1D9B overexpression than without (compare Figure 5F with Figure 6D).

*Response: We have conducted additional experiments to show that endogenous TBC1D9B undergoes complex formation with TMEM55B on lysosomes (**new Fig. 1h**). As TBC1D9B also binds to ARL8B-GTP in a tripartite complex - as demonstrated in the **new Fig. 5e** - it is plausible that its overexpression in fact may aid the recruitment of ARL8 to lysosomes under such artificial conditions.*

Overall, the data in the paper do not support many of the conclusions that the authors wish to make. It may be that ARL8B recruits TBC1D9B to lysosomes as an effector – perhaps to inactivate its known substrate Rab11, but the data do not support the conclusion that TMEM55B binds directly to TBC1D9B, nor the conclusion that TBC1D9B has GAP activity on ARL8B.

*Response: To address this suggestion by the referee, we have depleted cells of RAB11A/B or ARL8A/B to determine whether the effects of TBC1D9B loss on lysosome position are preserved under these conditions. We demonstrate in **the new Suppl. Fig. 4a,b** that TBC1D9B loss causes lysosome dispersion in WT and in RAB11A/B knockdown cells but not in cells depleted of ARL8B (as shown in main Fig. 7a,b). These data rule out the possibility that TBC1D9B exhibits its cellular effects indirectly via altering the nucleotide status of RAB11.*

Reviewer #3 (Remarks to the Author):

The manuscript 'Control of lysosome function by the GTPase-activating protein TBC1D9B and its binding partner TMEM55B' by Duhay and Tian et al, describes a number of interesting findings with regards to the role of the protein TBC1D9B in lysosomal function. The study identifies TBC1D9B as a key negative regulator of the lysosomal small GTPase ARL8b, showing that it binds ARL8b-GTP and promotes its inactivation. In addition, the study shows that loss of TBC1D9B, or its partner TMEM55B, disrupts lysosome positioning. The authors conclude that the loss of TBC1D9B affects autophagic flux, an effect that is dependent on ARL8b.

The finding that TBC1D9B has GAP activity towards ARL8b, is novel, and will be of interest to a broad community of researchers interested in organelle biology, membrane trafficking, and cellular nutrient sensing. Generally, the manuscript is well-written, the methodology is appropriate and most of the conclusions drawn are supported by the data.

Response: We thank the reviewer for the precise summary and these highly positive comments and his/ her support to publish our study in Nat Commun.

However, some conclusions are not supported by sufficient evidence and would require additional experiments to be conducted. It is recommended that the authors revise the current manuscript to address a few main concerns, as well as a list of minor items, appended below.

Response: We have addressed the few major and the list of minor points in the revised manuscript as explained below.

Main Items

1) Figure 3c-d shows an impaired increase in Cathepsin D activity in response to starvation in TBC1D9B and TMEM55B KO lines. However, it is not clear why this is, or even how the increased activity in WT cells is working. The paper cited (which is from one of the corresponding authors of this manuscript), Ebner M. et al. 2023, showed a lipid switch at lysosomes that alters lysosome functions including proteolytic activity, and blocking this lipid switch alters lysosome position. Another study also cited here, Johnson D.E. et al. 2016, shows that peripheral lysosomes are more alkaline – this could also stand to reduce Cathepsin D activity. Or, do the authors think that 6h starvation is causing significant transcriptional changes that boost Cathepsin D activity? The rationale and result should be discussed in more detail.

Response: We apologize for not explaining the assay and expected phenotype in detail. Briefly, we had shown in Ebner et al (2023) that during starvation, lysosomal cathepsin activity increases as a consequence of elevated PI(4)P production on lysosomes, which induces the recruitment of v1-ATPase and a resulting drop in lysosomal pH. This starvation-induced increase in lysosomal cathepsin activity is transcription-independent. We will explained this in more detail in the revised manuscript and discussed our results accordingly.

2) In the text relating to Figure 4, I would hesitate to conclude anything about lysosome “dynamics”. All the data is based on images of fixed cells. Dynamics implies that one would be measuring lysosomal speed or velocity, motility tracking to determine if they are undergoing motor-based transport, or lysosomal morphology changes/membrane remodeling such as tubulation, fusion, fission.

Live-cell imaging experiments would need to be performed to compute some of the above parameters and make conclusions about lysosome dynamics. Indeed, such additional data would strengthen the manuscript.

However, at the very least, Figure 4 (and the corresponding section in the text) should be labeled and

discussed as the effects on lysosome clustering/positioning upon overexpression of TBC1D9B and TMEM55B.

Response: We agree and have included live-cell imaging experiments by analyzing the kinetics of lysosome movement in WT and TBC1D9B loss-of-function models experimentally by additional live cell imaging experiments. These analyses revealed a significant increase in the fraction of moving lysosomes in both TMEM55B and TBC1D9B KO compared to wild-type cells (Figure 2e). Moreover, the average speed of lysosome movement was increased in TBC1D9B KO cells, a phenotype that was rescued by re-expression of TBC1D9B-eGFP (new Supplementary Figure 2a).

In Figure 4, we removed the term “dynamics” to avoid confusing the reader.

3) What is the knockdown efficiency of siTBC1D9B in Figure 7c-d? Western blot or, if there is no suitable antibody, qPCR, should be conducted and shown.

Response: We have quantified the efficacy of KD by immunoblotting and find TBC1D9B levels to be reduced to about 10% of those in control cells (see new Suppl. Figure 3d).

4) The single immunostaining experiment for LC3 is not sufficient to suggest that the lower levels of LC3 observed in the siTBC1D9B condition is due to increased autophagic flux as flux wasn't measured. An autophagy flux assay (for example, Halo-Tag Flux assay PMID: 36095089) should be performed, if the authors wish to make these conclusions.

Response: We thank the reviewer for this suggestion that we have addressed in two ways: First, we show that depletion of TBC1D9B results in a moderate accumulation of the ubiquitin adaptor and autophagic substrate p62 (new Suppl. Fig. 3e). In addition, we have used Halo-LC3 expressing cell lines kindly provided by Prof. Mizushima to monitor autophagic flux directly. These data, shown in the new Figure 3e, further confirm the reduction of autophagic flux assayed by auto-lysosomal cleavage of Halo-LC3.

On that note, how does this result (if the authors do suggest there is increased autophagic flux in siTBC1D9B cells) compare to the findings from Liao et al. 2018 that concluded TBC1D9B promoted autophagic flux and it's silencing reduced autophagic degradation?

Response: In the mentioned study by Liao et al (2018) it was shown that loss of TBC1D9B reduces the levels of LC3-II. These data are consistent with our results shown in Fig. 7c,d, in which we demonstrate that depletion of TBC1D9B causes a reduction of LC3 puncta in WT but not in ARL8A/B DKO cells.

Minor Items

-Regarding the callout of Figure 1f in the text: “Conversely, TBC1D9B-GFP was co-immunoprecipitated with TMEM55B (Figure 1f).” This second co-IP is in Figure 1e (right).

Response: Thanks, the mislabeling was corrected.

-Figure 1f – I would recommend moving all the red labels to one side (i.e. right) of the drawn constructs as it is a bit difficult, at first, to determine which label goes with which fragment

Response: We followed the suggestion and moved the red labels to the right.

-Supplemental Figure 1c: What is the asterisk in the HA blot referring to? Legend says the asterisk shows cleaved GFP, but it is also shown in the HA blot.

Response: The incorrectly inserted asterisk in the HA blot has been removed.

-Since OrgaMapper is used throughout to measure lysosome positioning, the authors should give a brief description (a couple of sentences) about how it works, in addition to the reference to the bioRxiv preprint that is already there. For example, readers will wonder if cell size is taken into consideration in this analysis since the TMEM55B and TBC1D9B KOs are obviously larger than WT. On

this note, this phenotype should be mentioned (if it is real), and followed up by a statement on how OrgaMapper normalizes organelle positioning to cell size (i.e. diameter?) and is therefore a suitable method. It would help the readers as they could have doubts about this and them having to go into the OrgaMapper preprint (as I did) to see whether this method controls for cell size, is a minor inconvenience.

Response: We updated the reference to the (now published) OrgaMapper paper. OrgaMapper measures both cell area and the cell's Feret's diameter (the longest distance between any two points in the cell's circumference). The reviewer is correct in his observation that the KO cells are larger. We now mention this fact in the main text to alert the reader. To account for these differences, all lysosome positioning data were computed using the average distance of lysosomes to the nucleus, normalized to the Feret's diameter, as pointed out in detail in the methods section.

-The statement: "Rab11-positive recycling endosomes were also partially redistributed to the cell periphery in TBC1D9B KO cells, consistent with an earlier study 34" should be accompanied by a callout to Supplemental Figure 2 (Rab11 staining)

Response: Thank you! We have added the reference to supplemental Figure 2a.

-Figure 2d should have a small legend for the schematic, as shown in Figure 4b. -Callout of Figure 2e in the text is missing a parenthesis

Response: Small legends were added to Figures 2d and former Figure 4d (now Fig. 4e) as suggested. The missing parenthesis was added.

-Figure 2e – The authors should show the individual channels in the whole-cell image. It looks, to this reviewer, like TBC1D9B-HA may have an ER localization, as well as some puncta that overlap with LAMP2. This ER localization is somewhat corroborated in Figure 2f, yet there is no discussion of this additional subcellular localization in cells. Is it also in the ER? If so, do the authors think this is real/expected? Or possibly a virtue of overexpression? An IF in which the ER is co-stained, would determine if any TBC1D9B signal is there. It is good that the overexpression is done in the background of the KO cells rather than WT to reduce overexpression-induced artefacts, however, TBC1D9B may be in the ER and this is not discussed. It is concluded that it has a cytosolic localization (and on some lysosomes) but the cytosolic localization is not convincing.

Response: We performed ER-co-staining of overexpressed TBC1D9B with the ER marker Calnexin and did not observe significant co-localisation between TBC1D9B and the ER.

Figure for referee #3:

-It is not obvious by eye in Figure 2f that the GAP activity-deficient TBC1D9B mutant is unable to rescue the lysosome dispersal phenotype. On the contrary, it looks like a partial rescue. Is this a matter of choosing a more representative image?

Response: We agree and have replaced the respective image for a more representative one.

-Figure 2g quantification should have the WT condition quantified as well because the experiment is measuring a “rescue” of lysosome positioning. This rescue can be best assessed by seeing the quantification of WT cells.

Response: Wildtype cells are included in the quantification.

-In this statement on pg 6: “As expected, starvation-induced marked relocalization of lysosomes towards the microtubule organizing center in wildtype cells.” there should be no hyphen, just “starvation induced...”

Response: The hyphen was removed.

-Is the TBC1D9B KO (fed) image in Figure 3a representative? The lysosomes do not look as peripheral as in Figure 2b and Figure 2f.

Response: Lysosome position is very heterogeneous between cells and, thus, individual images chose may not always represent the exact same distribution pattern. Even if the overall conditions for culturing cells are similar, parameters such as cell density and cell-to-cell variation will result in differential lysosomal distributions. This is why we take great care to quantify all data from many cells and across multiple independent experiments.

-Regarding Figure 3 c-d: The SiR lysosome signal in the fed condition, at least from the images in Figure 3c, doesn't seem to be the same between the genotypes (i.e. TMEM55B KO seems to have very diminished signal). Is this due to the extreme dispersal of lysosomes just making it difficult to see this signal by eye? SiR lysosome signal should be quantified in this condition alone across the different genotypes (fed) and perhaps placed into Supplemental.

*Response: The referee is correct that more dispersed lysosomes are more likely to display low fluorescence signals that make detection by eye difficult. Indeed, we found the SiR-Lysosomes levels in fed conditions alone are slightly different between the genotypes, however, these differences are not statistically significant (data now added as **new Supplementary Figure 3b**).*

-Supplemental Figure 3: Was the DQ-BSA allowed to co-endocytose with a pH- and protease-insensitive fluorescent probe, such as Alexa555-dextran to normalize for amount of endocytic uptake? It seems from the figure legend that cells were given DQ-BSA following the treatments (fed/starved). Thus, the difference in fluorescence intensity of DQ-BSA in the KOs could also be due to altered endocytosis. Authors are advised to either perform this experiment with an endocytosis control (as suggested above), or at the very least, mention this limitation of their current approach in the text.

*Response: We fully agree with the reviewer and added an experiment to test for the endocytic uptake of fluorescently labelled dextran under fed and starved conditions (**new Supplementary Figure 3c**). Under both conditions, no major differences in the endocytosis of dextran could be observed between the genotypes, suggesting that the differences in DQ-BSA hydrolysis are indeed due to proteolytic differences.*

-Figure 4c should be accompanied by the Pearson's correlation coefficient in TBC1D9B KO + TBC1D9B-HA staining with LAMP2. Since this result is not discussed, it is unclear if the Pearson's value shown in Figure 4c actually means loss of TBC1D9B from lysosomes and into the cytosol in the o/e condition. The result therefore seems arbitrary.

*Response: We included the Pearson's correlation coefficient in TBC1D9B KO + TBC1D9B-HA staining with LAMP2 in the figure (**new Figure 4d**).*

-It is not clear why Figure 5 a/b was done in iNeurons? It is also not clear why an “unbiased” approach needed to be taken here. The authors had just identified TBC1D9B as a regulator of lysosome positioning and it's GAP activity was important for this. The next logical step would be to look, in a targeted manner, at the lysosomal small GTPases known to couple to motors - ARL8 and Rab7. An explanation about the experimental rationale here would be useful.

Response: As the referee will note this is a collaborative study reporting on co-discoveries made by two laboratories independent of each other. The Haucke lab initially conducted proximity labeling proteomics to identify novel regulators of ARL8 function in axonal transport using iNeurons as a model. This work led to the identification of TBC1D9B as a GAP for ARL8 reported here. We have rephrased the manuscript to explain this coincident discovery that may be of interest to readers.

-In the statement "A GAP-activity-defective TBC1D9B mutant (TBC1D9BRYQ/AAA) (see Fig. 5a below) retained the ability to interact with ARL8b-GFP (Q75L) (Figure 5d)." What is the first callout "(see Fig 5a below)" referring to?

Response: The reference "(see Fig. 5a below)" was mislabeling and has been removed.

-Why does the overexpression of WT ARL8b (Figure 5f) show a lower Pearson's correlation coefficient for Lamp2/TBC1D9B? From the images it looks like there's good LAMP2/TBC1D9B overlap in the WT ARL8b condition.

Response: We thank the referee for noting this and have acted as follows: We have conducted additional experiments to increase cell numbers for analysis. In the revised figure 5f, we now show that overexpression of Arl8b-eGFP wild-type does not significantly impact TBC1D9B colocalization with lysosomal LAMP2, whereas overexpression of GTP-locked Arl8b causes the accumulation of TBC1D9B at lysosomes, consistent with our biochemical data. We have also replaced the representative images with new ones that align better with our quantitative analysis. The lack of effect of wild-type Arl8b on TBC1D9B levels at lysosomes likely reflects two effects: First, co-expressed TBC1D9B triggers GTP hydrolysis by Arl8b, resulting in its rapid inactivation and, thereby, loss of TBC1D9B binding. Second, the Pearson method only picks up comparably robust effects such as the those induced by overexpression of GTP-locked Arl8b. As the bulk of TBC1D9B and LAMP2 intensities remain perinuclear in Arl8b-overexpressing cells they dominate the Pearson's correlation over the small low-intensity peripheral lysosomes that arise from Arl8b overexpression.

-Figure 6d – The images for TMEM55B KO do not seem to show any more increase in ARL8b recruitment to LAMP2 compartments than WT. Are these images representative?

Response: We agree that the effect on the chosen images is not as clear as for the TBC1D9B KO, but these are representative images showing quite some ARL8B co-localising with LAMP2.

-In this statement on pg 9: " Given the defects of TBC1D9B-depleted cells with respect to the lysosomal starvation response (compare Figure 2), we monitored the presence of LC3 positive autophagosomes in cells depleted of TBC1D9B, ARL8a/b, or both proteins." In the callout, do the authors mean compare Figure 3?

Response: The reference to the figure has been corrected. We apologize for the error and thank the reviewer for noting the mistake.

-I am assuming from the discussion of Figure 7, that Figure 7c and d are showing cells under starvation condition? If so, this is not labeled in the micrographs (Fig 7c), the quantification (Fig 7d) or the figure legend. Is the quantification in Figure 7d referring to fold change in starved condition over fed condition? Is the WT siTBC1D9B quantification significantly different from WT scr? Also, check the labels in Figure 7d, they say TBC1B9D. And, as stated in main item #3, KD efficiency needs to be shown here.

Response: We have edited the corresponding figure legends to better explain these results. Moreover, we have quantified the efficacy of KD by immunoblotting and find TBC1D9B levels to be reduced to 10% of those in control cells (see new Suppl. Figure 3).

-With regards to this statement on pg 9: "Third, we show that TBC1D9B directly and selectively associates with active ARL8b-GTP in living cells and in vitro (Figure 4a-e)." Are the authors referring to Figure 5?

Response: The reference to the figure (Figure 5 instead of Figure 4) has been corrected. We apologize for the error and thank the reviewer for noting this.

-With regards to this statement on pg 10: "In preliminary experiments in HeLa cells overexpressing TMEM55B, we failed to detect any overt effects on TBC1D9B localization (Supplementary Figure 6a)." Supplemental Figure 6a doesn't show TBC1D9B staining, only LAMP2 and the TMEM55B-GFP expression in WT and TBC1D9B KO cells.

Response: Supplementary figure 6a has been corrected accordingly.

-In methods, change "Mutagenesis of TBCD9B" to "Mutagenesis of TBC1D9B"

Response: The typo was corrected.

Detailed response to the reviewers:

We were delighted to see that all three reviewers enthusiastically endorsed publication of our study in Nature Communications. We have addressed the remaining minor points as detailed below.

Reviewer #1 (Remarks to the Author):

The authors have addressed my comments sufficiently and the manuscript can be recommended for publication.

Reviewer #2 (Remarks to the Author):

This is a revised version of a paper that investigates the lysosomal membrane protein TMEM55B and finds an interaction with TBC1D9B which the authors then argue acts as a GAP for the lysosomal GTPase Arl8B, in addition to its previously reported role as a GAP for Rab11.

My review of the original version of the manuscript I noted the potential interest in the regulation of lysosomal positioning but expressed concerns about the strength of the evidence for TMEM55B binding directly to TBC1D9B and for the latter having GAP activity on Arl8B. In their rebuttal the authors state on several occasions that they "respectfully disagree" with my comments, but nonetheless they have added a significant amount of new data to strengthen these central conclusions of the paper. Overall, this has been done to a high standard and the paper seems significantly improved. As such I am happy to "respectfully" recommend acceptance. There are a couple of minor issues that would need addressing before publication:

Response: We thank the referee for these positive remarks and his/ her enthusiastic recommendation to publish our study in Nat. Commun.

a) The authors cite a recent study describing the binding between TMEM55B and a different partner, a study which synergises well with their own work. However, this paper is cited as Waschbusch et al 2025, but it is in fact from 2026.

Response: The paper has appeared in Structure in November 2025 as an "ahead of print" paper. It was now assigned to an issue (on February 5th 2026). We have updated the reference accordingly.

b) As the authors discuss, there are likely to be multiple regulators of Arl8B and its paralog Arl8A, and in this context they should cite a recent preprint that reports that DENND6A also has GAP activity toward ARL8B (PubMed ID 41542567).

Response: We added the mentioned preprint in the discussion:

*"A prediction from our work, therefore, is that other GAP proteins must exist that act to physiologically control ARL8 activity in cells and tissues. Such GAP proteins could conceivably act selectively on ARL8A or both ARL8 isoforms, or exhibit cell- and tissue-specificity. **Candidates are DENND6A, for which GAP activity was demonstrated in vitro (Vignogna & Fromme, bioRxiv 2026),** or other family members of the TBC family, including the TBC1D9B paralogs TBC1D9, TBC1D8, and TBC1D8B."*

Reviewer #3 (Remarks to the Author):

Overall, I believe the additional experiments and the corrections the authors have implemented have greatly improved the manuscript and I am satisfied with the authors' response. However, please correct all of the following:

Response: We would like to thank the reviewer for his/her positive feedback, careful reading and keen eye.

1) The manuscript file contains a duplication of the manuscript. Pages 1-36 is one manuscript without figures and pages 37 (labeled as page 1)-79 is another copy/version of the manuscript with figures. I assume this was a copy/paste error, but I am not sure if the two manuscripts are different in any way?

Response: This problem did not occur in the final version of the manuscript document. We assume that the submission system appended the manuscript versions with and without "track changes" to each other. In any case, the problem should no longer exist in the version that has now been uploaded.

2) Figure 2e and Supp Fig 2a – It is nice to see that lysosome speed has been computed. However, units of the speed are either incomplete (Figure 2f currently says "Speed>0.1 um", do the authors mean speed > 0.1 $\mu\text{m}/\text{sec}$?) or missing (Supp Figure 2 has no units of speed on y-axis or in the legend). Speed should be presented as distance/time.

Response: Thank you! We have updated the figure and corresponding legend accordingly.

3) Supp Fig 3d –The significance stars appear shifted to the left of the graph. The authors should check that this is moved to the right place. The legend for Supp Fig 3d doesn't correspond to TBC1D9B KD experiment.

Response: We thank the reviewer for noting this mistake, which has been corrected.

4) Supp Fig 3e – "TBC1D9B fed plus +BafA1", the authors may want to remove the "plus"? Again, the legend needs to be corrected for this panel.

Response: We would like to thank the reviewer for their careful reading and keen eye. The error has been corrected.

5) Supp Fig 3b – Legend needs correction – refers also to a dextran experiment.

Response: The legend for the Supplemental Figure 3 has been revised and corrected.

6) Overall for Supp Fig 3: Fix the entire legend as it also mentions a panel (f) and there is no panel f in the figure.

Response: The legend for the Supplemental Figure 3 has been revised and corrected.